# Comparing the performance of mScarlet-I, mRuby3, and mCherry as FRET acceptors for mNeonGreen

**Tyler W. McCullock**, **David M. MacLean, Paul J. Kammermeier***

Department of Pharmacology and Physiology, University of Rochester Medical Center, Rochester, New York, United States of America

* paul_kammermeier@urmc.rochester.edu

**Data Availability Statement:** All relevant data are within the manuscript and its Supporting Information files.

## Abstract

Förster Resonance Energy Transfer (FRET) has become an immensely powerful tool to profile intra- and inter-molecular interactions. Through fusion of genetically encoded fluorescent proteins (FPs) researchers have been able to detect protein oligomerization, receptor activation, and protein translocation among other biophysical phenomena. Recently, two bright monomeric red fluorescent proteins, mRuby3 and mScarlet-I, have been developed. These proteins offer much improved physical properties compared to previous generations of monomeric red FPs that should help facilitate more general adoption of Green/Red FRET. Here we assess the ability of these two proteins, along with mCherry, to act as a FRET acceptor for the bright, monomeric, green-yellow FP mNeonGreen using intensiometric FRET and 2-photon Fluorescent Lifetime Imaging Microscopy (FLIM) FRET techniques. We first determined that mNeonGreen was a stable donor for 2-photon FLIM experiments under a variety of imaging conditions. We then tested the red FP's ability to act as FRET acceptors using mNeonGreen-Red FP tandem construct. With these constructs we found that mScarlet-I and mCherry are able to efficiently FRET with mNeonGreen in spectroscopic and FLIM FRET. In contrast, mNeonGreen and mRuby3 FRET with a much lower efficiency than predicted in these same assays. We explore possible explanations for this poor performance and determine mRuby3's protein maturation properties are a major contributor. Overall, we find that mNeonGreen is an excellent FRET donor, and both mCherry and mScarlet-I, but not mRuby3, act as practical FRET acceptors, with the brighter mScarlet-I out performing mCherry in intensiometric studies, but mCherry out performing mScarlet-I in instances where consistent efficiency in a population is critical.

## Introduction

Genetically encoded Fluorescent Proteins (FPs) have advanced basic and translational biology immensely. Starting with the cloning of the *Aequorea victoria* green FP[1], a massive and continual effort to expand the number of available FPs with a different physical and spectral properties began. Currently, there is an enormous variety of FPs at all parts of the visible spectrum,

**Funding:** This project was supported by National Institute of Health (https://www.nih.gov/) funding R00NS094761 and Brain and Behavior Research Foundation NARSAD Young Investigator Award (https://www.bbrfoundation.org/grants-prizes/narsad-young-investigator-grants) to D.M.M. and a University of Rochester (https://www.rochester.edu/) Interim Funding award to P.J.K.. T.W.M. was supported in part by the University of Rochester John R. Murlin Fund. The funders had no role in study design, data collection and analysis, decision to publish, or preparation of the manuscript.

**Competing interests:** The authors have declared that no competing interests exist.

and even some parts of the ultraviolet and infrared spectrums. These new proteins were either cloned from other organisms[2–4], or developed through evolution of already identified FPs [4–11]. This ever expanding catalog of FPs has been reviewed by others[12–14], and new efforts and archives have been created to organize this information, such as the FPbase database[15].

As the catalogue of FPs has expanded, so have the potential uses. Of particular note is the use of FPs as biosensors which can measure signaling events, cell metabolites, pH, voltage and more[16]. Many of these biosensors employ Förster Resonance Energy Transfer (FRET) as part of their reporting mechanism, producing a change in acceptor emission upon donor excitation when the quantity of interest changes. FRET is also orientation and distance dependent [17], meaning that the magnitude of energy transfer (described as the FRET efficiency) can be used to study either the steady state or dynamic changes in protein interactions. Much of the work utilizing FPs for FRET has been done using Cyan/Yellow FP pairs due to their brightness, but these proteins suffer from large overlaps in their emission spectra making interpretation of results unnecessarily challenging, among other complicating factors. Green/Red FP pairs offer greater separation of their emission spectra while maintaining the high degree of spectral overlap between donor emission and acceptor absorption that allows for efficient energy transfer. Green/Red FPs also offer greater Förster radii than most Cyan/Yellow protein pairs, enabling FRET over longer distances. Being able to detect FRET over larger distances often leads to greater dynamic range and sensitivity. Additionally, cellular toxicity to blue light has been well documented in a variety of systems[18–22], emphasizing the need for better green and red shifted FRET pairs. Historically, Green/Red FRET has been limited by unfavorable fluorescent properties of the red protein[23–26], but several recently developed monomeric red fluorescent proteins are reported to have improved absorption, brightness, and stability indicating they may act as high quality FRET acceptors for green FPs.

It is the aim of this study to characterize these recently released red FPs as FRET acceptors in hopes of helping aid a more general adoption of Green/Red FRET. Here, we investigate the ability of next generation red FPs, mRuby3 and mScarlet-I as well as mCherry to act as a FRET acceptor for the green-yellow FP donor mNeonGreen. mRuby3[10] is the newest iteration of the Ruby series of red FPs[8, 9] originally developed from eqFP611[3]. The mScarlet series of red FPs[11] was developed from a synthetic gene template based off several naturally occurring red FPs. Three monomeric red FPs were evolved from this strategy, the bright variant mScarlet, the fast maturing variant mScarlet-I, and the fast lifetime variant mScarlet-H. For this study, we use mScarlet-I due to its fast maturation time, as slower maturation is less ideal for fusion proteins studies. mCherry[6], a commonly used monomeric red FP developed from DsRed[2], is included in this study to act as a standard. For the donor molecule, we chose mNeonGreen[4], an incredibly bright and stable green-yellow FP derived from the monomerization of the tetrameric yellow fluorescent protein LanYFP[4].

## Materials and methods

### Cell culture and transfection

HEK293A cells (ATCC; Manassas, VA) were maintained in Dulbecco's Modified Eagle Medium (DMEM) supplemented with 1X GlutaMAX™, 100 units/mL penicillium, 100 μg/mL streptomycin (ThermoFisher Scientific; Waltham, MA), and 10% heat-inactivated fetal bovine serum (Atlas Biologicals; Fort Collins, CO). To induce expression of fluorescent proteins, cells were transiently transfected using polyethyenimine (PEI) transfection. For all transfections the indicated amounts of DNA and PEI where mixed together in DMEM containing no supplements for 30 minutes before addition to cells at the indicated time point. For FLIM

experiments, cells were plated on number 1 cover slips (Warner Instruments; Hamden, CT) in 12-well plates (Nest Scientific USA; Rahway, NJ) 48 hours prior to experiments. Cells were then transfected 24 hours before experiments began using 0.6 µg of cDNA for the indicated construct and 1.6 µg of PEI in 1 mL of media. For confocal experiments, cells were plated in 35 mm plastic culture dishes (Corning; Corning, NY) 48–72 hours prior to experiments, followed by transfection with 1 µg of cDNA using 2.4 µg of PEI in 2 mL of media 24–36 hours before imaging began. For spectral FRET and immunoblotting experiments, cells were plated in 10 cm plastic culture dishes and allowed to grow to 70% confluency. Cells were then transfected with 4 µg of cDNA using 7.8 µg of PEI 24 hours before experiments started.

## Plasmids and cloning

All constructs used in this manuscript are derived from the pKanCMV-mClover3-mRuby3 plasmid (Plasmid #74252) available on Addgene.org (Addgene; Watertown, MA). NG-mRuby3 was created by removing mClover3 and all but the last 5 amino acids of the linker from pKanCMV-mClover3-mRuby3 by inverse PCR, and the full mNeonGreen coding sequence was inserted in its place using In-Fusion cloning (Takara Bio; Mountain View, CA). NG-mScarlet-I, NG-mScarlet, and NG-mCherry were created by deleting mRuby3 from NG-mRuby3 by inverse PCR and insertion of either mScarlet-I, mScarlet, or mCherry using In-Fusion cloning (Takara Bio). NG-Stop was created using the same inverse PCR product as NG-mScarlet-I and NG-mCherry via blunt end ligation using NEBs KLD Mix (New England Biolabs; Ipswich, MA). NG-P2A-mRuby3 was derived from the NG-mRuby3 using inverse PCR and blunt end ligation to insert the P2A sequence between the linker and mRuby3. The mRuby3-GGSGG-NG construct was made using a multi-fragment In-Fusion reaction using mRuby3 and mNeonGreen PCR products where the GGSGG linker was appended to the 3' end of mRuby3 and the 5' end of the mNeonGreen product via PCR. mClover3-Stop was made by deleting mRuby3 from the pKanCMV-mClover3-mRuby3 vector by inverse PCR and subsequent blunt end ligation. The amino acid sequences for all constructs described here is available in S1 Table. The mNeonGreen gene was obtained from the pmNeonGreen-NT plasmid (Allele Biotechnology; San Diego, CA), the mScarlet-I gene was obtained from the Lck-mScarlet-I plasmid (Addgene, plasmid #98821), and the mCherry gene was obtained from the pmCherry-N1 plasmid (Takara Bio). Reaction products were transformed, screened, and amplified in XL10-Gold Ultracompetent *E. coli* cells (Agilent Technologies; Santa Clara, CA). Plasmid were isolated using an E.Z.N.A Plasmid MiniPrep Kit (Omega Bio-teck; Norcross, GA) or a ZymoPure II Plasmid Midiprep Kit (Zymo Research; Irvine, CA) per manufactures recommendations. All constructs were verified by sanger sequencing (Eurofins Genomics; Louisville, KY) and stored at a concentration of 1µg/µL in a -20˚C freezer. All plasmids and constructs generated for this manuscript are freely available upon request.

## Spectral FRET

For spectral FRET experiments, cells were used 24–36 hours post transfection. A single 10 cm dish of cells expressing the construct of interest were washed with imaging buffer (136 mM NaCl, 560 µM $MgCl_2$, 4.7 mM KCl, 1 mM $Na_2HPO_4$, 1.2 mM $CaCl_2$, 10 mM HEPES, 5.5 mM Glucose) several times and pelleted. Cell pellets were then resuspended in 500 µL of imaging buffer and transferred to disposable acrylate cuvettes (Spectrocell Inc; Oreland, PA). Emission scans were collected from 490–750 nm using a 470 nm excitation wavelength using a Cary Eclipse Fluorescence Spectrophotometer (Agilent Technologies). Cell were resuspended by vigorous pipetting immediately prior to scans. Data was collected in the Cary Eclipse Scan Application software package (Agilent Technologies) and exported to Microsoft Excel

(Microsoft Corporation) for analysis. To estimate FRET efficiency from emission scans, linear unmixing was performed in Igor Pro (WaveMetrics Inc; Lake Oswego OR) using donor only and acceptor only emission scans to determine the contributions of donor and acceptor. Due to the high level of consistency between the NG-Stop and reported spectrum for purified mNeonGreen (S1 Fig), the purified mNeonGreen spectrum was used as the donor only scan for linear unmixing. For each of the acceptors, custom acceptor only scans were created based on experimentally collected data and their reported purified spectrum (as discussed in S1 Fig). All raw scans used for spectral FRET are available in S1 Fig. Once the contributing weight of donor and acceptor was determined, FRET efficiency was estimated using the following equation[27]:

$$E = \frac{W_A}{\left(\frac{QY_A}{QY_D}\right)W_D + W_A} \; x \; 100\%$$

(1)

where E is the FRET efficiency, $W_A$ and $W_D$ are the component weights of the donor only and acceptor only emissions calculated through linear unmixing, and $QY_A$ and $QY_D$ is the quantum yield of the acceptor and donor respectively.

## Fluorescent lifetime imaging microscopy (FLIM)

For FLIM experiments, cells were used 1-day post transfection unless otherwise noted. 30 minutes before experiments begin, media was exchanged for imaging buffer and allowed to come to room temperature. Coverslips were transferred to a custom-built chamber and mounted onto the stage of an Olympus FluoView 1000-MP Multiphoton and Scanning Confocal Microscope (Olympus; Tokyo, Japan). Excitation of samples was achieved using a Mai Tai Ti:Sapphire Femtosecond Pulse Laser (Spectra-Physics; Santa Clara, CA) tuned to 950 nm. Fluorescent decays were collected using a XLPlan N 25x (NA 1.05) water immersion objective (Olympus) at 256x256 resolution and a pixel dwell time of 20 µsec. The data was passed by the microscope to the FLIM set up, consisting of two H7422P Hamamatsu detectors (Hamamatsu; Hamamatsu City, Japan) and a time-correlated single photon counting card (Becker and Hickl; Berlin, Germany). To separate donor and acceptor emission, a BrightLine FF552-Bi02-25x36 dichroic (Semrock; Rochester, NY) was used. Emission light to the donor detector was further filtered with a BrightLine FF01-510/42-25 filter and emission to the acceptor detector was filtered with a BrightLine FF01-609/54-25 filter (Semrock). The microscope was controlled with the FV10-ASW software package (Olympus) and the FLIM system was controlled via the VistaVision by ISS software package (ISS Inc; Champaign, IL). Analysis of single cell lifetime decays was performed with VistaVision by ISS and subsequent lifetimes were analyzed in Microsoft Excel (Microsoft Corporation; Redmond, WA). Briefly, cells were isolated into individual regions of interest (ROIs), and the decay data for each pixel in the region above threshold were summed to create a cell fluorescence decay curve. These decay curves were then fit and lifetimes were extracted using non-linear regression in tail-fitting mode. Almost all decay curves were well described by a single exponential fit (as determined by Chi-squared analysis as well as by eye). Lifetimes were extracted from these fits and used to calculate FRET efficiency using the following equation[28]:

$$E = 1 - \left(\frac{\tau_{DA}}{\tau_D}\right)$$

(2)

Where E is the FRET efficiency, $\tau_{DA}$ is the lifetime of the donor in the presence of acceptor, and $\tau_D$ is the lifetime of the donor only species (in this case, the average lifetime of NG-Stop).

For experiments where acceptor photobleaching was preformed, cells were bleached by scanning with a 559 nm laser at 100% power for 15 minutes under the same parameters that the lifetime data was collected.

## Confocal microscopy

For confocal microscopy experiments, cells were used 1-day post transfection. 30 minutes before experiments began, media was exchanged for imaging buffer and allowed to come to room temperature. Images were then collected on the same microscope as the FLIM data utilizing a SIM scanner and 488 nm and 559 nm conventional laser lines. Widefield confocal images were collected through a XLUMPlanFl 20X (NA 0.95) water immersion objective (Olympus) at a resolution of 2048x2048. The sample was excited, and emission collected in each channel individually to prevent blead through. Green emission was collected through a 505–540 nm filter and red emission was collected through a 575–675 nm filter. All imaging parameters were kept consistent across experiments to facilitate comparisons. Data was collected in FV10-ASW software package (Olympus) and exported to the Fiji software package [29] for analysis as described in the main text and S3 Fig using the Coloc2 plugin.

## Immunoblotting

Samples for immunoblotting were harvested 24 hours after transfection. Cells were removed from their culture dish and washed twice with calcium and magnesium free DPBS (Corning). Following cell lyses in the presence of protease inhbitors, protein concentrations were evaluated using a Pierce$^{TM}$ BCA Protein Assay Kit (ThermoFisher Scientific) and 10 μg of total protein was loaded into a 16% SDS-PAGE gel. After electrophoresis, proteins were transferred to a 0.2 μm nitrocellulose membrane (BioRad; Hercules, CA). Presence of the mNeonGreen protein was probed using a monoclonal mouse anti-mNeonGreen antibody [32F6] (Chromotek; Planegg-Martinsried, Germany) at a 1:1000 dilution and visualized using a donkey anti-mouse IgG DyLight 680 secondary antibody (ThermoFisher Scientific) at a 1:5000 dilution. The same membrane was additionally probed for the presence of mClover3 using a rabbit anti-GFP antibody (ThermoFisher Scientific, A11122) at a 1:1000 dilution and visualized with a goat anti-rabbit IgG DyLight 800 secondary antibody (ThermoFisher Scientific) at a 1:10000 dilution. Detection was completed using an Odyssey 3 Imaging System (Li-cor Biosciences; Lincoln, NE).

## Statistics

To avoid assumptions regarding the data distributions, all statistical significance in this manuscript was determined using a permutation test implemented through a custom Python script utilizing the MLxtend library[30] using the approximation method with 1,000,000 permutations. For instances where there was less than 10 data points total between the two data sets being compared, an exact method was used.

# Results

## Physical properties and spectrum of proteins in this study

For this study, we utilized the monomeric, yellow-green fluorescent protein mNeonGreen to test the ability of two new red fluorescent proteins, mRuby3 and mScarlet-I, as well as mCherry to act as FRET acceptors. The fluorescent and physical properties of these proteins are listed in Table 1. mNeonGreen is one of the brightest fluorescent proteins to date with a high quantum yield and extinction coefficient making its signals easy to observe. In addition,

**Table 1. Properties of the fluorescent proteins used in this study.**

| Protein | Ref | $\lambda_{ex}$ max (nm) | $\lambda_{em}$ max (nm) | EC ($M^{-1}cm^{-1}$) | QY | pKa | Maturation (min) | $J(\lambda)$ ($\times 10^{15}$ $M^{-1}cm^{-1}nm^4$) | $R_0$ (Å) |
|---|---|---|---|---|---|---|---|---|---|
| **Donors** | | | | | | | | | |
| mNeonGreen | 4 | 506 | 517 | 116,000 | 0.8 | 5.7 | 10 | - | - |
| mClover3 | 10 | 506 | 518 | 109,000 | 0.78 | 6.5 | 43.5 | - | - |
| **Acceptors** | | | | | | | | | |
| mRuby3 | 10 | 558 | 592 | 128,000 | 0.45 | 4.8 | 136.5 | 4.65 | 64 |
| mScarlet-I | 11 | 569 | 593 | 104,000 | 0.54 | 5.4 | 36 | 3.60 | 61 |
| mCherry | 6 | 587 | 610 | 72,000 | 0.22 | 4.5 | 15 | 2.28 | 57 |
| mScarlet | 11 | 569 | 594 | 100,000 | 0.7 | 5.3 | 174 | 3.51 | 61 |

$\lambda_{ex}$ max = emission maximum, $\lambda_{em}$ = emission maximum, EC = extinction coefficient, QY = quantum yield, $J(\lambda)$ = overlap integral, $R_0$ = Förster Radius. $J(\lambda)$ and $R_0$ are calculated with mNeonGreen as the donor, an orientation factor of 0.6667, and a refractive index of 1.33. Values were obtained from the references listed under the Ref column.

mNeonGreen is also highly photostable[4], and matures quickly, making it an ideal FRET donor. Both mRuby3 and mScarlet-I have high extinction coefficients, poising them to be excellent FRET acceptors. In comparison to mCherry, these proteins also offer substantially higher quantum yield, indicating that it will be easier to detect energy transfer events using the emission of one of these two proteins. The absorbance and emission spectrum of mNeonGreen is overlaid with the spectrum of mRuby3, mScarlet-I and mCherry in Fig 1A–1C. For each red fluorescent protein, the overlap integral ($J(\lambda)$) and Förster radius ($R_0$) values with mNeon-Green are listed in Table 1. Based on these photophysical properties, all three red FPs are expected to FRET with mNeonGreen. Assuming identical positioning between donor and acceptor, mRuby3 is expected to produce the highest FRET efficiency and mCherry the lowest FRET efficiency with mNeonGreen.

## mNeonGreen-RFP tandem constructs reveal poor performance of mRuby3

To test the ability of the red FPs to act as FRET acceptors, we constructed tandem FP constructs consisting of the full coding sequence of mNeonGreen (NG) followed by a short amino

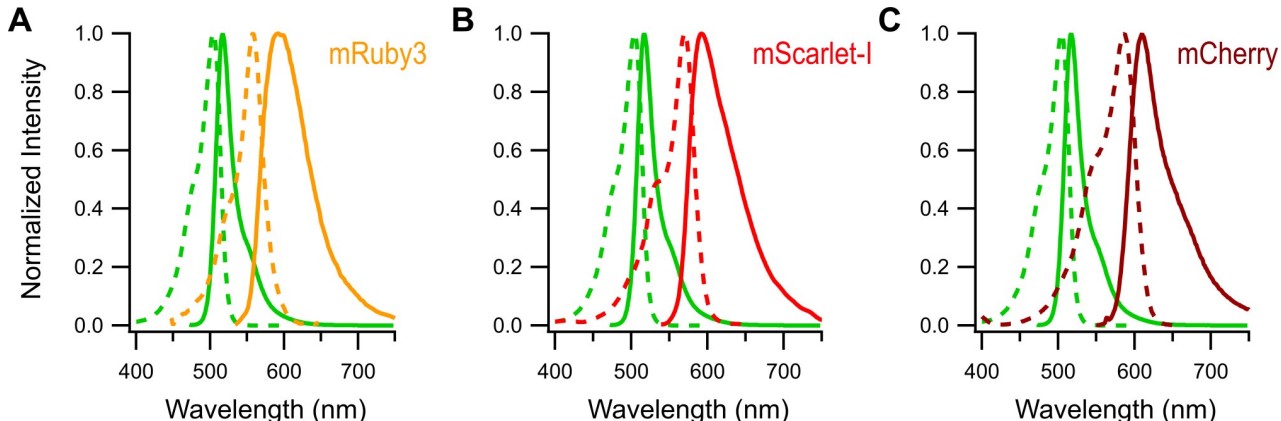

**Fig 1. Spectrum of mNeonGreen, mRuby3, mScarlet-I and mCherry.** Absorbance (dashed lines) and emission spectrum (solid lines) of purified mNeonGreen (green lines) overlaid with the spectrum of purified mRuby3 (**A**), mScarlet-I (**B**), and mCherry (**C**). Spectrum were obtained from the reference indicated in Table 1.

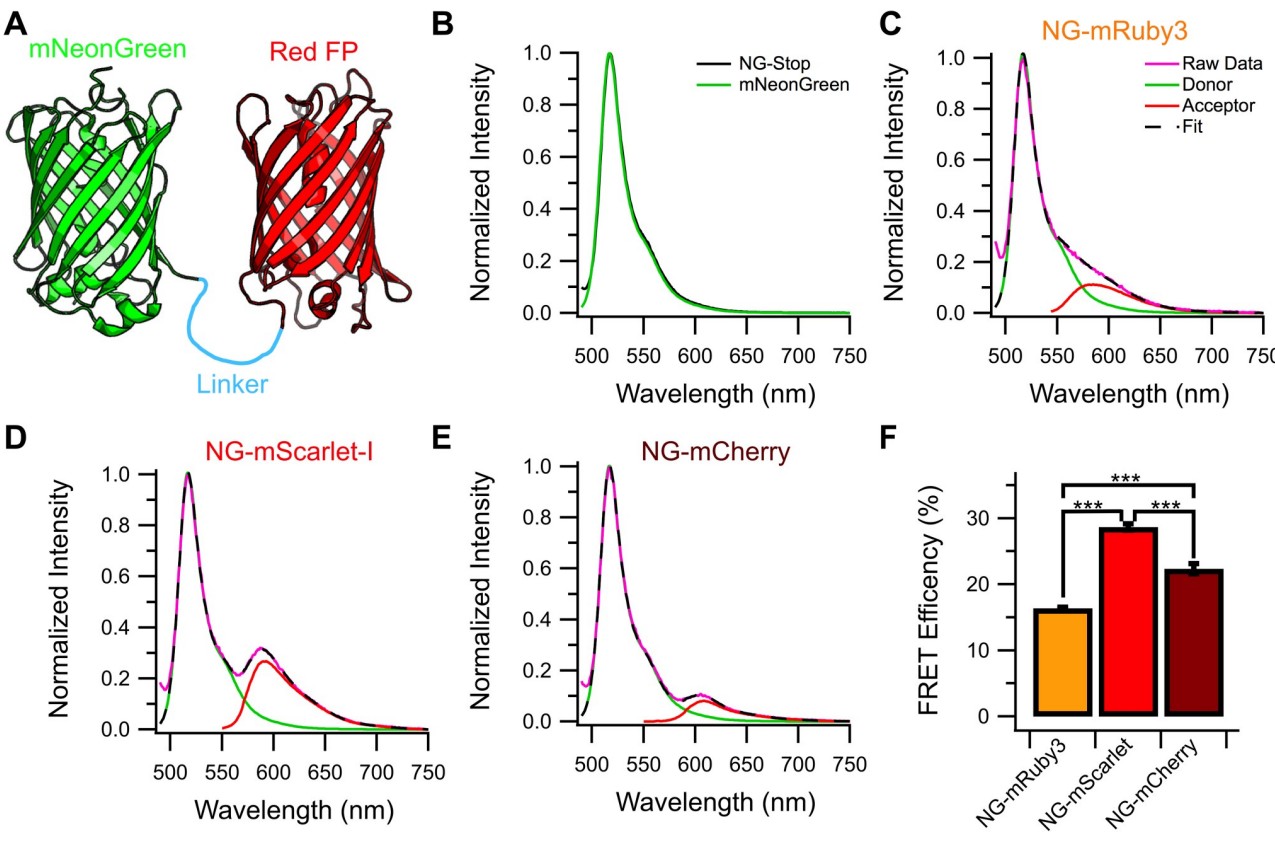

**Fig 2. Spectral FRET of each mNeonGreen-Red FP tandem constructs. (A)** Cartoon schematic of the mNeonGreen-Red FP tandem constructs used for FRET experiments. **(B)** Average emission scan of cells expressing NG-Stop (black) when excited at 470 nm overlaid with the reported spectrum for purified mNeonGreen in green (n = 3 independent transfections). Example raw emission spectrum (pink) of tandem **(C)** NG-mRuby3, **(D)** NG-mScarlet-I, and **(E)** NG-mCherry when excited at 470 nm. The dashed black line shows the sum of donor (green) and acceptor (red) components calculated by linear unmixing. **(F)** FRET efficiencies calculated from the spectrum for each construct (n = 3). *** = P < 0.0005 between the indicated conditions.

acid linker (SGLSKGEE) and then the full coding sequence of the red FP (Fig 2A with the amino acid sequences available in S1 Table in the Supplemental Information). Although an 8 amino acid linker is relatively short, inclusion of the full mNeonGreen c-terminus (which appears unstructured or absent in the mNeonGreen structure[31]) creates an effective linker of 17 amino acids. A mNeonGreen-Stop construct was also created, which contains the full coding sequence of mNeonGreen, the 8 amino acid linker, and then a stop codon to act as a donor only control construct. To determine if mNeonGreen will FRET with the red FPs in these constructs, we transiently transfected each construct into HEK293 cells and measured the acceptor FP emission upon excitation with a wavelength that will only excite the donor. When cells expressing the NG-Stop constructs are excited with 470 nm light, a spectrum comparable to what has been reported for purified mNeonGreen was obtained (Fig 2B, S1 Fig in the Supplemental Information). When the NG-red FP tandem constructs are assayed, a second peak emerges corresponding to the red FP. Example spectrum for the NG-mRuby3, NG-mScarlet-I, and NG-mCherry constructs are shown in Fig 2C, 2D and 2E respectively in pink, along with calculated donor (green) and acceptor components (red) determined through linear unmixing, as well as the fit (black dashed line) resulting from the addition calculated donor and acceptor components. Note that in all cases, the unmixing fit faithfully reproduced the raw data traces. Using the donor and acceptor components, the FRET efficiency of each

construct can be estimated using Eq. 1 (Fig 2F, with the raw traces available in S1 Fig). Over three independent transfections for each construct, the NG-mScarlet-I and NG-mCherry construct presented a FRET efficiency of 29 ± 0.5% and 22 ± 0.8% respectively. Surprisingly, the NG-mRuby3 construct presented an estimated FRET efficiency of 16 ± 0.1%. This poor performance in comparison to NG-mScarlet-I or NG-mCherry was highly unexpected considering the physical and spectral properties reported for mRuby3[10], which predicted it would be the best FRET acceptor of the three red FPs for mNeonGreen.

## mNeonGreen is a suitable donor for two-photon FLIM

To confirm our spectral FRET findings, we turned to the more precise method of 2-Photon Fluorescent Lifetime Imaging (FLIM). We first sought to determine if mNeonGreen was a suitable donor for 2-Photon FLIM experiments. A well-behaved FRET donor would be easily excited by the 2-photon laser, be stable under a variety of 2-photon laser powers, and display a stable, mono-exponential lifetime[32]. To our knowledge, there is only one report of mNeon-Green's performance using two-photon illumination[33]. This study demonstrated that blue shifted fluorescent proteins tend to perform better under 2-photon excitation than more yellow shifted fluorescent proteins but did not preclude mNeonGreen's use from two-photon based studies. Using 950 nm light and the NG-Stop construct, we assayed mNeonGreen's performance in 2-photon FLIM over various conditions using Time Correlated Single Photon Counting. Over a variety of laser powers, mNeonGreen produced a stable lifetime of 3.05 ± 0.01 ns over 50 frames of acquisition (Fig 3A). This lifetime was stable for up to 300 frames of acquisition for all but the highest laser power tested, 25 W/cm$^2$ in which the lifetime linearly decayed to 87 ± 4.5% of its initial value after 300 frames (Fig 3B). In contrast, significant photobleaching was observed for laser powers above 15 W/cm$^2$ (Fig 3C) with 20 W/cm$^2$ and 25 W/cm$^2$ bleaching approximately 48 ± 14% and 94 ± 3% of the sample intensity respectively after 300 frames. Example decay traces of a single cell collected in consecutive 50 frame intervals are shown at 15 W/cm$^2$ (Fig 3D) and 25 W/cm$^2$ (Fig 3E). Normalizing the decay traces to the peak intensity for the 25 W/cm$^2$ example demonstrates the reduction in lifetime and the decay in data quality over the course of acquisition at this power (Fig 2F). To serve as a comparison, we repeated these experiments using EGFP (S2 Fig). EGFP produced a stable lifetime of 2.63 ± 0.01 ns over various laser powers. Similar to mNeonGreen, this lifetime was stable for 300 frames of acquisitions for each laser power other than 25 W/cm$^2$. Additionally, power levels above 15 W/cm$^2$ also induced significant photobleaching over 300 frames, although to a lesser degree than what was observed with mNeonGreen (S2 Fig). Given these data, we concluded that mNeonGreen is a suitable donor for two-photon FLIM experiments when the lifetime is acquired at a laser power of 15 W/cm$^2$ or less. For all further FLIM experiments, data was collected at laser powers between 5–10 W/cm$^2$ for 100–150 frames depending on the brightness of the cell.

## FLIM-FRET measurements confirm poor performance of mRuby3

With FLIM, FRET is detected as a reduction of the donor's lifetime when in the presence of the acceptor[28]. This technique has the advantage of determining the FRET state between mNeonGreen and the red FP independent of the red FPs emission, allowing us to determine the FRET efficiency for each construct without having to account for the differences between each red FPs. Over the course of several independent transfections, the mNG-Stop construct produced a donor only lifetime of 3.05 ± 0.02 ns (Fig 4A). When mScarlet-I or mCherry are present in the tandems, mNeonGreen's lifetime is reduced to 2.22 ± 0.06 ns and 2.23 ± 0.03 ns respectively (Fig 4A). This results in a FRET efficiency of the NG-mScarlet-I construct of

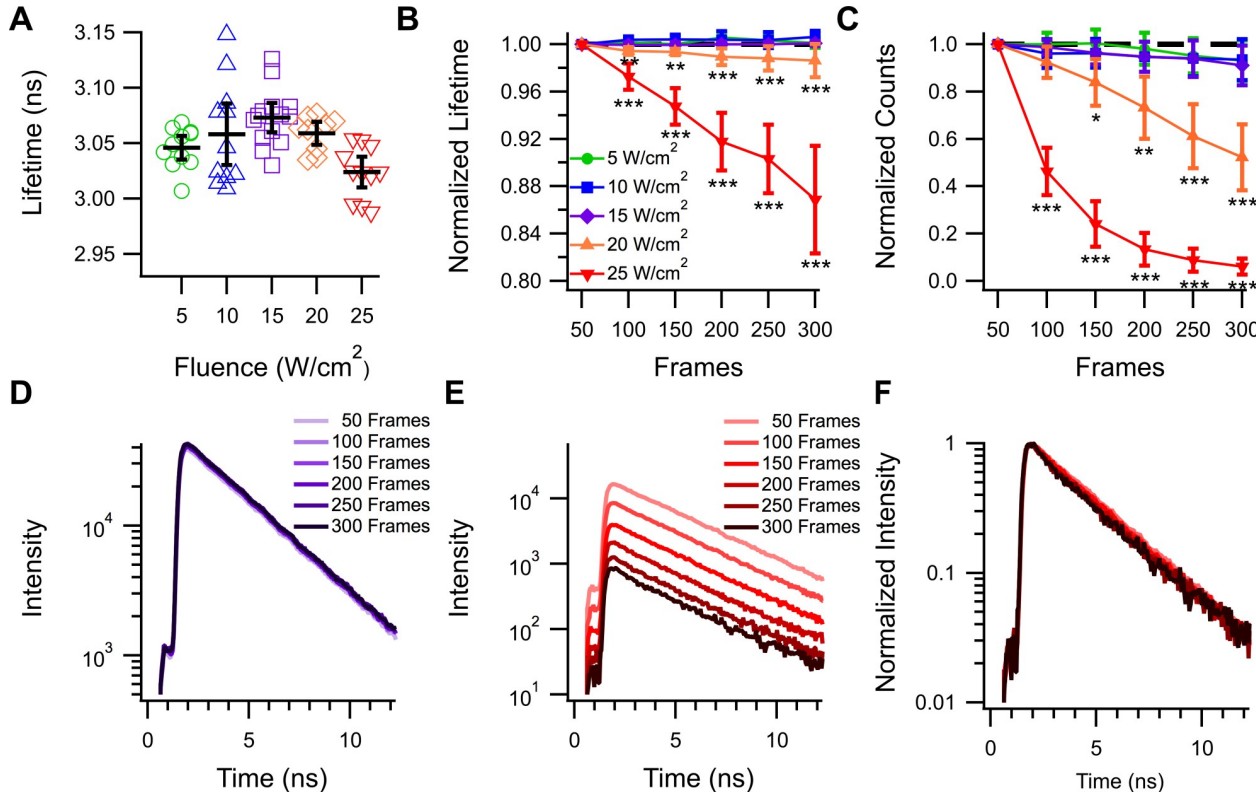

**Fig 3. mNeonGreen performs well under various laser powers for two-photon time domain FLIM acquisitions.** (A) Lifetime data collected from individual HEK293 cells expressing cytosolic mNeonGreen at various laser powers up to 25 W/cm$^2$ after 50 frames. Black bars indicate the average ± 95% confidence interval. **(B)** Lifetime and **(C)** intensity of samples taken over 300 frames at various laser powers. $^* = P < 0.05$, $^{**} = P < 0.005$, and $^{***} = P < 0.0005$ compared to the frame matched 5W/cm$^2$ dataset. N for each sample is as follows 5W/cm$^2$: 11 cells, 10W/cm$^2$: 10 cells, 15W/cm$^2$: 14 cells, 20W/cm$^2$: 9 cells, 25W/cm$^2$: 11 cells. **(D)** Example lifetime decay curves obtained at 15 W/cm$^2$ over 300 frames. **(E)** Example lifetime decay and normalized decays **(F)** obtained at 25 W/cm$^2$ over 300 frames.

27 ± 2% and an efficiency of 27 ± 1% for the NG-mCherry construct (Fig 4B). Assay of the NG-mRuby3 construct shows even worse performance of mRuby3 than what was estimated with spectral FRET (Fig 2F). The average lifetime of the NG-mRuby3 construct was 2.92 ± 0.03 ns (Fig 4A) resulting in a FRET efficiency of only 4 ± 0.9% (Fig 4B). Indeed, 35 cells across 3 independent transfections (over half of all cells sampled) exhibited lifetimes within the range of what was collected for the NG-Stop construct, whereas no cells expressing NG-mScarlet-I or NG-mCherry had lifetimes within that range (Fig 4A). Example decay traces representative of the average for each construct are shown in Fig 4C and example lifetime maps of HEK293 cells expressing each construct are shown in Fig 4D.

## Confocal imaging demonstrates lower than expected red fluorescence for mNeonGreen-mRuby3 construct

To further investigate the poor performance of mRuby3 as a FRET acceptor, we turned to confocal microscopy to analyze the behavior each tandem construct. The presence of mNeon-Green and the red FPs were independently surveyed via sequential excitation with a 488 nm and 559 nm lasers, with the emissions of each channel being collected separately to prevent bleed through. Example wide field images of cells expressing each construct are shown in Fig 5A. Qualitatively, it can be seen in the merged image of the green and red channels that the

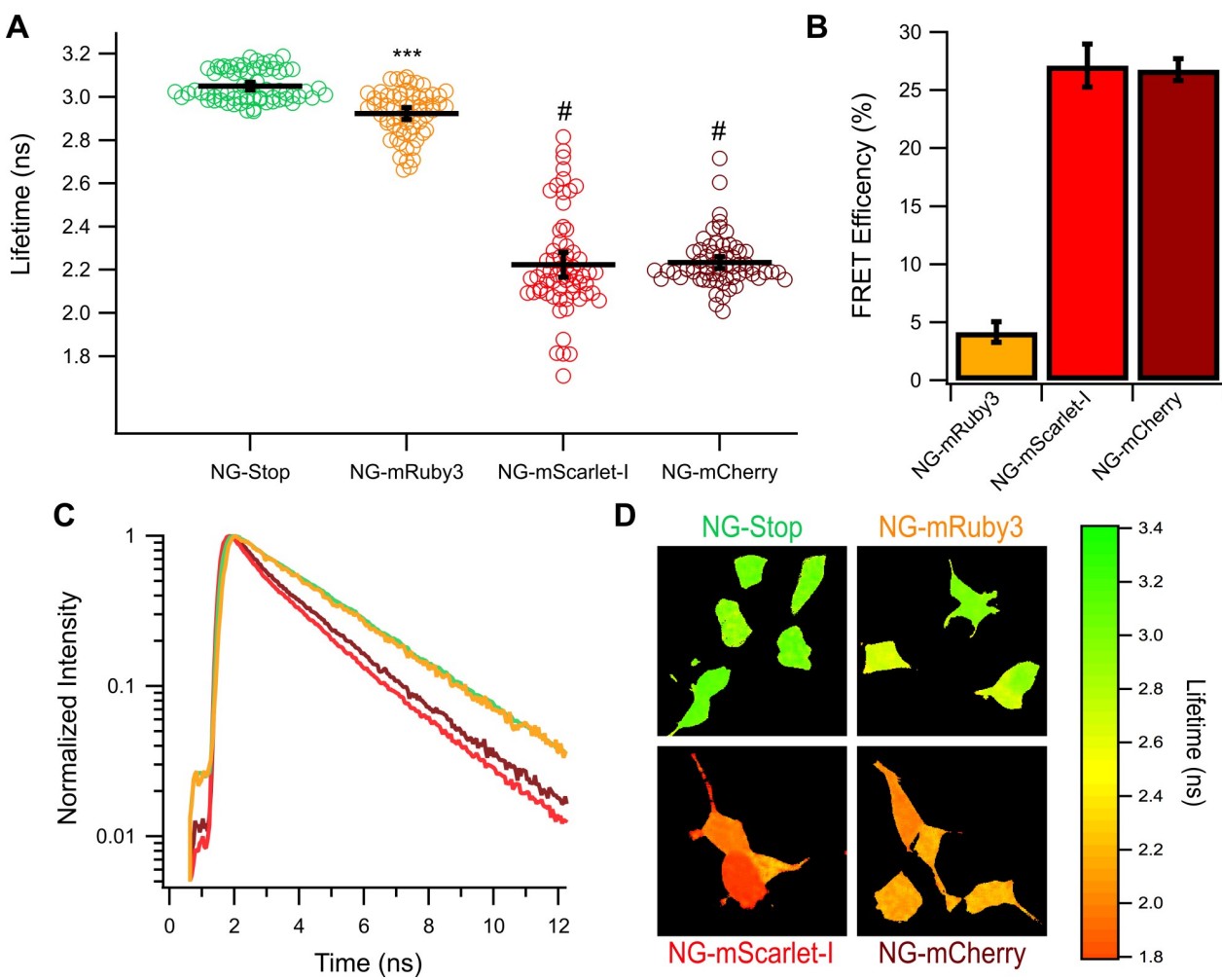

**Fig 4. Lifetime of mNeonGreen-Red FP tandem constructs. (A)** The lifetime of mNeonGreen in individual cells expressing a mNeonGreen-Red Protein tandem fusion construct. Black bars indicate the average ± 95% confidence interval. **(B)** FRET efficiency calculations for each tandem construct. *** = P < 0.0005 compared to NG-Stop and # indicates P < 0.0005 compared to NG-Stop and P < 0.0005 compared to NG-mRuby3. NG-mScarlet-I and NG-mCherry are not statistically different (P = 0.75) **(C)** Example decay curves for each tandem representative of the average lifetime of all cells for each construct. **(D)** Example lifetime heat maps for a single frame for each construct. N for each construct is as follows NG-Stop: 68 cells, NG-mRuby3: 63 cells, NG-mScarlet-I: 64 cells, and NG-mCherry: 64 cells.

cells expressing the NG-mRuby3 construct had widely varying intensities of green and red fluorescence. Interestingly, cells were regularly observed that seem to contain high levels of mNeonGreen and low levels of mRuby3 or vice versa, alongside cells that appeared to contain both proteins. This heterogeneity was also noted in with the NG-mScarlet-I and NG-mCherry constructs, but with much less frequency (Fig 5A). This result was quite surprising given the constructs were designed to express mNeonGreen and the red FP stoichiometrically and in tandem. Theoretically, we would expect expression of such tandem constructs to result in a fixed ratio of green to red intensity where the red intensity varies depending on the brightness of the red FP and the green intensity varying due to changes in the extent of FRET. To test this prediction and quantify the heterogeneity for each construct we examined regions of interest (ROI) containing a single cell, extracted the green and red intensities for that cell following background subtraction, and plotted the resulting intensities at the pixel level. The resulting green-red intensity plots for each cell were then fit with a linear regression. An example of this

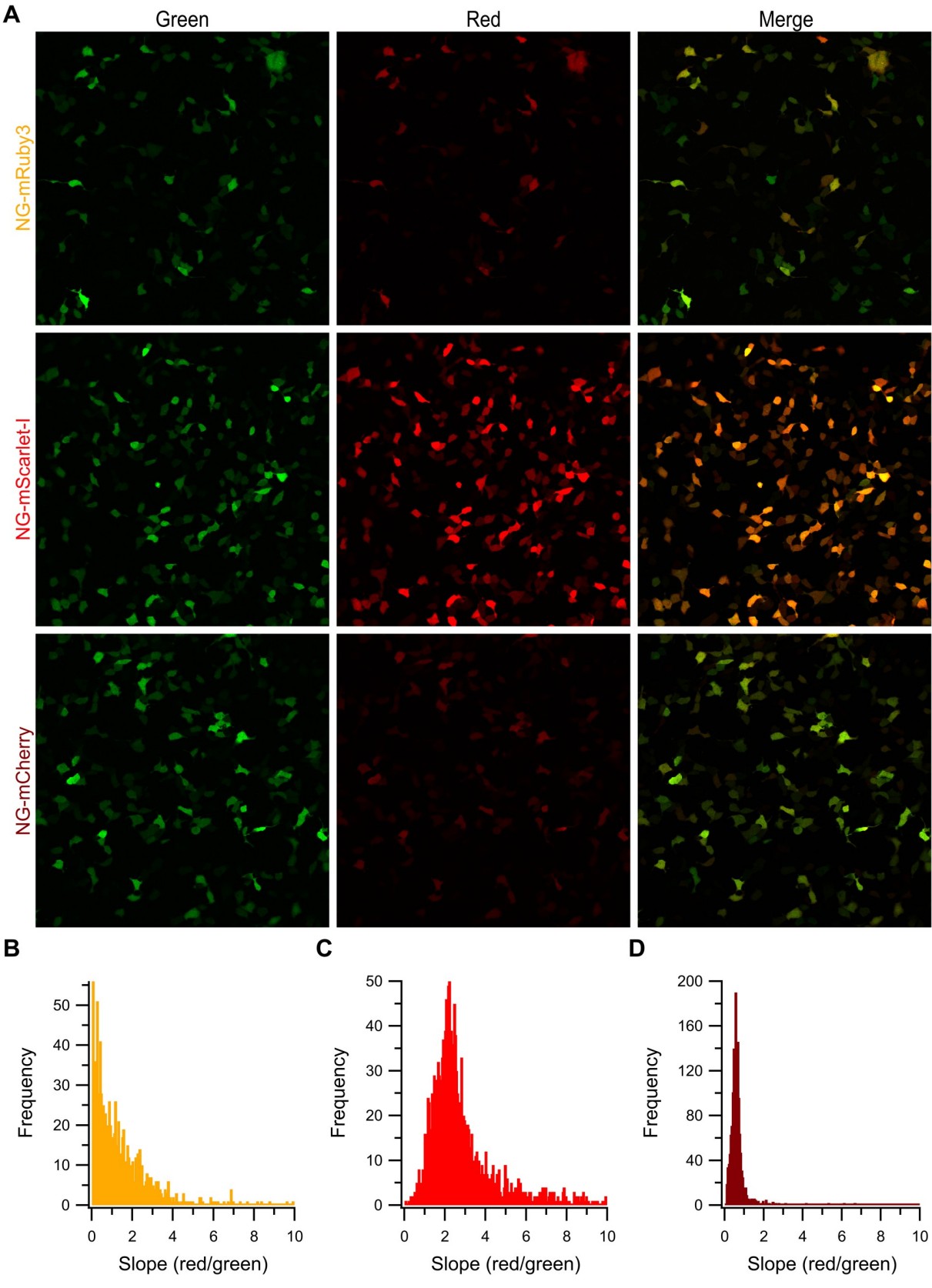

**Fig 5. Confocal microscopy of the tandem constructs. (A)** Example images of NG-Red FP constructs when directly excited by 499 nm or 559 nm lasers. Histograms of the slopes of the red/green intensity correlations for individual cells expressing **(B)** NG-mRuby3 (n = 1077 cells), **(C)** NG-mScarlet-I (n = 1745 cells), and **(D)** NG-mCherry (n = 1557 cells).

workflow is diagrammed in S3 Fig. While the absolute value of this slope will be different for each fluorophore pair and for different imaging conditions, the changes in this slope from cell to cell under the same imaging conditions will report the heterogeneity of each construct. As expected of a tandem construct and based on qualitative assessments of images (Fig 5A), histograms of the slopes for individual cell expressing the NG-mCherry and NG-mScarlet-I constructs revealed a reasonable distribution and narrow spread of slopes (Fig 5C and 5D). This indicates the majority of cells expressing these constructs contain a fixed ratio of mNeonGreen to red FP. In contrast, the histogram of slopes from cells expressing NG-mRuby3 were not as evenly distributed (Fig 5B). Indeed, the majority of cells expressing this construct have a slope close to 0, suggesting that these cells contain measurable mNeonGreen levels but low levels of fluorescent mRuby3. Interestingly, this lack of abundant red fluorescence does not stem from a lack of mRuby3 protein as western blotting of cells transfected with NG-mRuby3 produce a similar banding pattern as those transfected with NG-mScarlet-I and NG-mCherry (S4 Fig). Importantly, in NG-mRuby3 transfected cells there is no smaller monomer sized band, indicating our tandem constructs are generally intact and ruling out the possibility of mNeonGreen being produced without mRuby3. Taken together, these data provide a mechanistic explanation for our FLIM data in which all cells expressing the NG-mScarlet-I and NG-mCherry constructs exhibited robust FRET, whereas a large majority of NG-mRuby3 constructs did not. Further, they suggest that the poor performance of NG-mRuby3 may be due to poor maturation of the acceptor compared to NG-mScarlet-I and even NG-mCherry.

## mRuby3 exhibits similar behavior in several configurations and similar behavior to the slow maturing mScarlet variant

To better determine if mRuby3's behavior was specific to our system, or intrinsic to mRuby3 itself, we developed several additional tandem constructs. First, we swapped the orientation of the tandem to see if mRuby3 was more tolerant of c-terminal fusions and changed the linker to a slightly smaller glycine-serine linker (GGSGG). This mRuby3-GGSGG-NG construct preformed similarly the original NG-mRuby3 construct, producing an average lifetime of 2.96 ± 0.02 ns (Fig 6A) corresponding to a FRET efficiency of 4 ± 1% (Fig 6B). Confocal analysis of this construct (Fig 6C and 6D) also reveals a similar phenotype to NG-mRuby3, where the majority of cells have slopes close to 0, indicating most cells have detectable mNeonGreen fluorescence, but lack mRuby3 fluorescence. Next, we inserted an optimized self-splicing peptide sequence (P2A peptide)[34] between NG and mRuby3 in the NG-Ruby3 construct. This sequence causes a ribosomal skipping event during translation that will result in the release of the NG polypeptide from the ribosome during translation, before re-initiating translating the mRuby3 portion of the construct, resulting in two separate proteins being translated from the same transcript. Notably, we designed the construct so that most of the extra P2A peptide sequence will be appended to NG, with the only addition to mRuby3 being an N-terminal proline upon successful cleavage. Cells expressing NG-P2A-mRuby3 exhibited an average lifetime of 3.09 ± 0.01 ns (Fig 6A) and displayed lifetimes that were statistically indistinguishable from the lifetimes of cells expressing NG-Stop, collected on the same day. This result was expected as the NG and mRuby3 in these cells should no longer be linked together, and indeed, western blot analysis of this construct confirms the P2A peptide is cleaving with high efficiency (S4 Fig). Unfortunately, though this construct shows only a minor improvements in the behavior

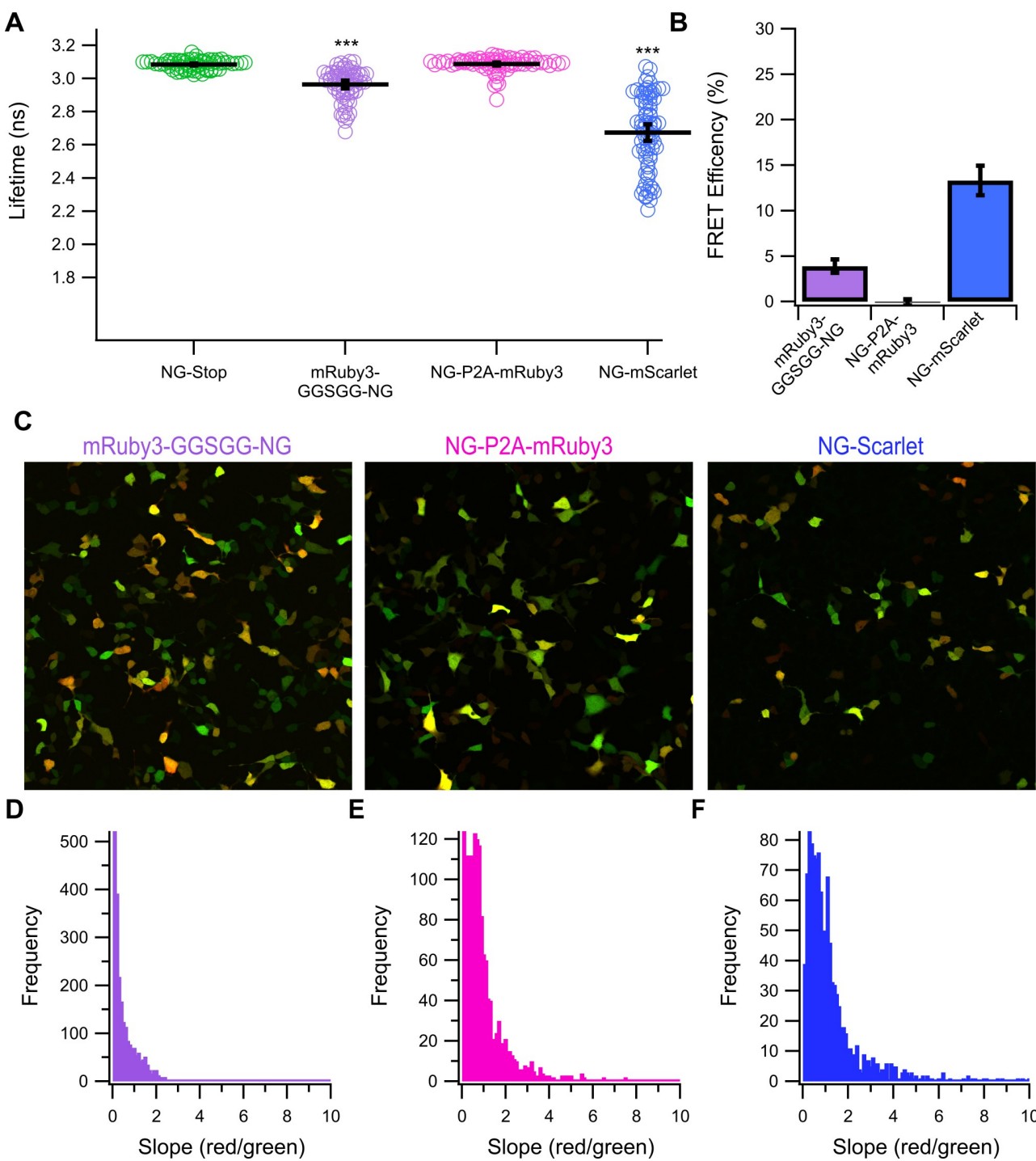

**Fig 6. Behavior of distinct mRuby3 and mScarlet containing tandem constructs. (A)** The lifetimes of a distinct set of NG-Stop express cells with the lifetimes of 2 new mRuby3 containing tandem constructs and an mScarlet containing construct. Black bars indicate the average ± 95% confidence interval. *** = P < 0.0005 compared to NG-Stop. NG-Stop and NG-P2A-mRuby3 are not statistically different (P = 0.65). N for the lifetime measurements of each construct is as follows NG-Stop: 64 cells, mRuby3-GGSGG-NG: 66 cells, NG-P2A-mRuby3: 69 cells, and NG-mScarlet: 77 cells. **(B)** Average FRET efficiency for each of the new tandem constructs. **(C)** Example merge confocal images of HEK293 cells expressing each of the new tandem constructs. Pooled histograms of the slopes of cells expressing **(D)** mRuby3-GGSGG-NG, **(E)** NG-P2A-mRuby3, and **(F)** NG-mScarlet. N for the confocal measurements of each construct is as follows mRuby3-GGSGG-NG: 2302 cells, NG-P2A-mRuby3: 1548 cells, and NG-mScarlet: 1150 cells.

of mRuby3 as determined by confocal analysis (Fig 6C and 6E), which still showed a high portion of cells with near zero slopes, but did also exhibit a secondary, non-zero peak.

While mScarlet-I and mCherry mature with a half-time of 36 and 15 minutes respectively, mRuby3 is reported as having a maturation half-time of 136.5 minutes. To test if the large difference in maturation times could result in the phenotype we see for mRuby3, we mutated the mScarlet-I in the NG-Scarlet-I tandem to the bright but slow maturing Scarlet variant, mScarlet, by introducing a I74T mutation. This slow maturing variant is reported as having a maturation half-time of 132.4 minutes under the same conditions as the mRuby3 maturation measurements were made. Cells expressing NG-mScarlet presented highly variable lifetimes averaging to 2.67 ± 0.05 ns resulting in an average FRET efficiency of 13 ± 2% (Fig 6A and 6B). Under confocal, cells expressing this NG-Scarlet also display a heterogeneous phenotype (Fig 6C) but, a majority of cells exhibit a non-zero slope indicating that most cells exhibit both green and red fluorescence (Fig 6F).

Lastly, we turned to analyzing mRuby3's behavior with an evolutionarily distinct green FP, mClover3. Using a mClover3-mRuby3 tandem construct available from Addgene, we created a mClover3-Stop construct to serve as a donor only control and analyzed the ability of mRuby3 to act as an acceptor for mClover3 using FLIM. In addition to using a different green FP, this construct also uses a 12 amino acid linker rather than the effective 17 or 15 amino acid linkers we have used thus far (S1 Table). Cells expressing mClover3-Stop exhibit a stable monoexponential lifetime of 3.11 ± 0.01 ns (S5 Fig). When mRuby3 is present in the construct to act as an acceptor, this lifetime is reduced to 2.81 ± 0.05 ns on average (S5 Fig). This results in an average FRET efficiency of 10 ± 2%. It is also worth noting that similar to NG-mScarlet, mClover3-mRuby3 exhibited a large range of lifetimes. Given that mClover3's spectral and physical properties are very similar to mNeonGreen's, this seems to indicate a slight improvement in performance on mRuby3's behalf, either due to the sorter linker, or the inclusion of mClover3 over mNeonGreen. Regardless though, a high heterogeneity is still seen with the expression of this tandem (S5 Fig) indicating the underlying problems with mRuby3 are still present.

## Expressing mNeonGreen-mRuby3 for longer periods of time improves its performance

Our previous data has suggested that mRuby3's poor acceptor properties may in part be due to inefficient maturation. To test this further, we performed FLIM experiments for up to five days following transient transfection with the NG-mRuby3 construct. As seen in Fig 7A, the average lifetime of mNeonGreen in the NG-mRuby3 construct decreases from 2.92 ± 0.03 ns 1-day post transfection (DPT) to 2.41 ± 0.09 ns 5 DPT. This results in a change in average FRET efficiency of 4 ± 0.9% 1 DPT to an efficiency of 21 ± 3% efficiency 5 DPT transfection (Fig 7B). Example lifetime maps of individual cells expressing NG-mRuby3 2–5 DPT are shown in Fig 7C, with the example lifetime map for 1 DPT shown in Fig 4D. Example fluorescent decay traces for each DPT are available in S6 Fig. As time progressed, the number of cells exhibiting lifetimes within the range of the NG-Stop also decreased. However, even 5 DPT cells could still be observed that exhibited donor only like lifetimes. This indicates that although inefficient maturation in mammalian cells may be part of the reason mRuby3 performs poorly in previous experiments, it is likely not the only contributing factor. To ensure that the changes observed over time for the NG-mRuby3 construct were due to changes occurring to the mRuby3 protein, and not mNeonGreen, mNeonGreen lifetimes of the NG-mRuby3 constructs were determined before and after acceptor photobleaching on 5 DPT (Fig 7D). Regardless of the lifetime each cell exhibited before acceptor photobleaching, all cells

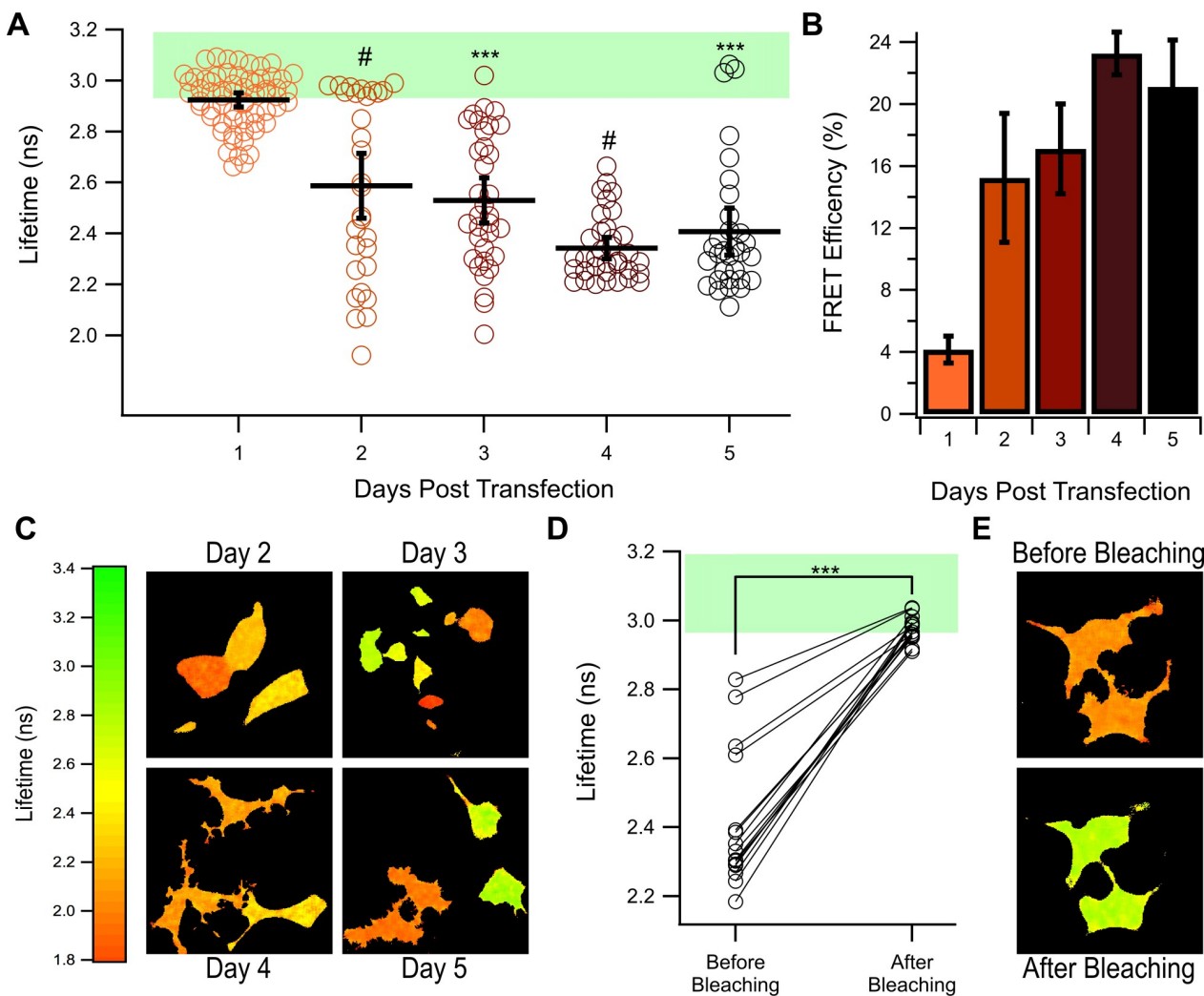

**Fig 7. FRET of NG-mRuby3 1–5 days post transfection. (A)** Lifetimes of cells expressing NG-mRuby3 construct 1–5 days post transfection (DPT) with 1 DPT is replicated from Fig 3A for reference. Black bars indicated the average lifetime ± 95% confidence interval. The green shading indicates the range of lifetimes observed from the NG-Stop construct. *** = P < 0.0005 compared to 1 DPT, and # = P < 0.0005 compared to 1 DPT and P < 0.0005 compared to the day before. N for each condition is as follows, 2 DPT: 30 cells, 3 DPT: 34 cells, 4 DPT: 35 cells, and 5 DPT: 31 cells. **(B)** Average FRET efficiency of NG-mRuby3 1–5 DPT. **(C)** Example lifetime maps collected each day tested. **(D)** Lifetime data from NG-mRuby3 expressing cells 5 days post transfection before and after acceptor photobleaching (n = 15). *** = P < 0.0005 after photobleaching compared to before photobleaching. **(E)** Example lifetime maps of the same cells before and after acceptor photobleaching.

exhibited lifetimes similar to that of the NG-Stop construct after acceptor photobleaching. Fig 7F shows example lifetime maps of two cells expressing NG-mRuby3 5 days post transfection before and after acceptor photobleaching and example fluorescent decay traces can be found in S6 Fig. These findings demonstrate that the mNeonGreen lifetime remained stable over the course of 5 days, indicating that the changes the NG-mRuby3 construct underwent was due to changes occurring to the mRuby3 protein.

## Discussion

Development and verification of bright monomeric Green/Red FP pairs will greatly increase the general adoption of Green/Red FRET. Bright fluorophores allow for easier detection and greater signal to noise ratio. Using green and red fluorophores specifically has several distinct

advantages include reduced toxicity by the excitation source, greater spectral separation, larger Förster radii, and better tissue penetrance of excitation and emission wavelengths. These benefits allow for newer, more accurate and precise studies to be conducted with less confounding factors than what could be done with Cyan/Yellow FRET pairs. mNeonGreen is an ideal donor for Green/Red FRET experiments. Its yellow shifted excitation and emission spectrum allow for a high degree of overlap with red FPs while also being capable of being excited with lower energy blue light than cyan or more blue shifted green FPs. In addition, mNeonGreen is remarkably bright under single-photon illumination, making its signals easy to observe. Both mScarlet-I and mRuby3 are reported as having some of the highest extinction coefficients of any red FPs, as well as also being some of the brightest monomeric red FPs to date. These facts indicate that they should make excellent FRET acceptors, making Green/Red FRET more accessible to researchers. In this study, we aimed to test this prediction using a variety of optical techniques.

We tested the ability of these two new red FPs–mRuby3 and mScarlet-I–along with mCherry to act as FRET acceptors for mNeonGreen using a tandem protein approach. Initially we used intensiometric spectral FRET to estimate the FRET efficiency between mNeonGreen and the red FPs (Fig 2). In this assay NG-mScarlet-I demonstrated the highest FRET efficiency, followed by NG-mCherry, and then NG-mRuby3 (Fig 2F). This was surprising given the reported properties of mRuby3 predicted that it would be the best acceptor for mNeonGreen. Specifically, mRuby3 had the highest degree of overlap between its excitation spectrum and mNeonGreen's emission spectrum and it has the highest extinction coefficient of the three red FPs in this study (Table 1). These experiments also demonstrated the advantages of these newer proteins over mCherry, as mScarlet-I produced almost 3 times the intensity of mCherry, despite only a 7% difference in estimated efficiency, and mRuby3 produced almost 1.5 times the intensity of mCherry, despite NG-mRuby3 exhibiting a lower FRET efficiency than NG-mCherry. This reflects the difference in extinction coefficient and quantum yield between these two proteins and mCherry and demonstrates how mScarlet-I is a better overall acceptor that will provide more signal and greater dynamic range for intensiometric green-red based FRET experiments.

To confirm these results, we turned to a more precise technique, FLIM. FLIM has the advantages of only needing to observe only the donor, there for eliminating the need to correct our data for differences in the acceptors physical properties as reported by others. In addition, our FLIM set up allows for assaying single cells to allow for a more detailed profile of the tandem construct's behavior. After establishing mNeonGreen as a suitable donor for 2-Photon FLIM experiments (Fig 3), we used this technique to measure the FRET efficiency of each tandem in the study (Fig 4). We found that both mScarlet-I and mCherry were able to efficiently FRET with mNeonGreen as almost to the same degree, but that mRuby3 was unable to induce substantial FRET (Fig 4A and 4B). It was somewhat surprising to see mCherry preformed just as well as mScarlet-I as a FRET acceptor in our FLIM experiments, despite having a significantly lower extinction coefficient. This may be a demonstration of mCherry's fast maturation, which may allow for the highest possible amount of fully developed tandems. To further investigate mRuby3s poor performance, we employed confocal microscopy to reveal great heterogeneity in the way the NG-mRuby3 tandem expresses (Fig 5A and 5B), with many cells exhibiting either red or green fluorescence, without the other. This was in spite of western blotting analysis suggesting that most cells should be expressing the full-length tandem protein (S4 Fig). This behavior was consistent across three additional mRuby3 constructs (Fig 6). Changing the orientation of the constructs and shortening the linker between them (Ruby3-GGSGG-NG) seemed to have virtually no effect on mRuby3's performance as an acceptor, or the observed heterogeneity under confocal (Fig 6). Only minor improvements were observed under confocal with the

NG-P2A-mRuby3 construct which would allow mRuby3 to mature without being covalently linked to NG (a conformation which prevented FRET but allowed us to assess mRuby3's behavior as an independent protein). Notably, this behavior was similar to what was observed with the slow maturing Scarlet variant, mScarlet (Fig 6), suggesting mRuby3's poor performance may be due to its longer maturation time. Given these findings, were conducted experiments on the NG-mRuby3 construct several days following transfection and indeed found that allowing the NG-mRuby3 construct more time to mature improved its performance as a FRET acceptor (Fig 7).

In theory, the higher extinction coefficient and quantum yield of mRuby3 predicts that it should outperform the mCherry and mScarlet-I. Shockingly though, we found the opposite result (Fig 2 and Fig 4). Even under the most ideal conditions tested (4 or 5 days post transfection), the NG-mRuby3 tandem still did not reach the FRET efficiencies of the NG-mScarlet-I and NG-mCherry tandems (Fig 7). This is in contrast to a previous report of a similar mNeon-Green-mRuby3 construct preforming very well, achieving nearly 40% efficiency in both HEK293 and Hela cells[10]. At this point, the reasons for this discrepancy is unclear, although similarities in linker length and composition, transfection times, and cell types suggest it is not due to a difference between the makeup of the tandem constructs used or differences in the conditions in which the experiments were conducted (for what conditions were reported). This inconsistency also followed through for the mClover3-mRuby3 construct (S5 Fig), once again, for reasons unknown. In addition, we found during our experiments that the emission spectrum of mRuby3 in this tandem was slightly left shifted by 6 nm compared to what is reported purified mRuby3[10] (S1 Fig). This finding was not reported previously and although this shift was slight, it was necessary to correct for it in order to properly fit our data. The cause of this spectral shift remains unclear but may simply be a result of mRuby3 having slightly different behavior in cells verse in a purified system. This may also be due to the photochromic behavior previously reported for mRuby3[11]. Whatever the cause, unpredictable changes in spectrum are highly concerning for FRET experiments as they can result in changes in fluorescence intensity that may be misinterpreted as changes in FRET state. It is possible that that this behavior also influenced our spectral FRET experiments (Fig 2) and could account for the disparity between the calculated FRET efficiencies for the NG-mRuby3 construct in the spectral FRET vs. FLIM assays.

While few studies using mRuby3 have been published to date, studies utilizing its predecessor, mRuby2 report generally positive results[4, 9–11, 35, 36]. Although, a similar study to ours using the green FP mClover as a donor reported much lower than expected efficiencies when using mRuby2 as an acceptor[37]. A commonality between our study and theirs is the use of mammalian cells lines. This coupled with our data reported here suggest that poor or slow maturation efficiency of the Ruby series is a major contributor to mRuby3's poor performance 1–3 days post transfection. Issues of maturation could also compound with variability in protein turnover rates from cell to cell or protein stability issues, and a combination of these factors likely explains the high degree of heterogeneity we observe under confocal with our mRuby3 constructs as well as with the slow maturing variant of mScarlet (Fig 5 and Fig 6).

Overall we find that both mCherry and mScarlet-I act as excellent acceptors for mNeon-Green within our model system, with each protein offering distinct advantages depending on the detection systems being utilized. Additionally, we demonstrated how factors beyond classically considered FRET parameters (donor quantum yield, spectral overlap, and acceptor extinction coefficient) can affect FRET systems in unpredictable ways. It was one of these "non-classic" factors, maturation time, that causes us to conclude that use of mRuby3 (and possibly the slow maturing mScarlet) should only be done with extreme caution where the drawbacks we experienced (heterogenous expression and poor performance shortly after

expression begins) will be mitigated or outweighed by potential advantages of mRuby3. It is also important to consider that each of these proteins may behave differently in different systems. This remains a possible explanation for the differences between our data and that reported by others, and our conflicting data underscores the importance of well thought out testing and controls particularly when developing more complex FRET system.

## Conclusion

In this study, we test the viability of mNeonGreen as a FRET donor, and mRuby3, mScarlet-I, and mCherry as FRET acceptors with a tandem FP model system. We find that mNeonGreen performs well as a FRET donor in both intensiometric and 2-photon FLIM experiments. When testing the red FPs as acceptors for mNeonGreen, we found that both mScarlet-I and mCherry were readily able to FRET with mNeonGreen. These proteins preformed equally well in FLIM experiments, but mScarlet-I outperformed mCherry in the intensiometric study due to its higher quantum yield. However, when FRET detection is measured independent of acceptor emission intensity, such as in our FLIM system, mCherry demonstrates equal FRET efficiency but more consistent expression, making it a superior choice over mScarlet-I in this scenario. In contrast, we find that mRuby3 performs poorly as a FRET acceptor in our model system, despite it being predicted to be the best FRET acceptor of the three proteins. A major contributor to this poor performance is mRuby3's slow maturation, although protein stability, and cell-to-cell heterogeneity in protein turnover may also contribute or compound this problem.

Overall, we found that mNeonGreen makes an excellent green-yellow donor, and mScarlet-I is one of the best all-around acceptors for green-red FRET experiments utilizing fluorescent proteins.

## Supporting information

**S1 Fig. Construct emission scans used for spectral FRET. (A)** Emission scans of several independent transfections of NG-Stop when excited at 470 nm overlaid with the reported emission of pure mNeonGreen. The average of these scans is shown in Fig 2B. Because of the consistency of NG-Stop with the pure mNeonGreen spectrum, the pure mNeonGreen spectrum was used as the donor spectrum for linear unmixing. Acceptor emission scans from several independent transfections of **(B)** NG-mRuby3, **(D)** NG-mScarlet-I, and **(F)** NG-mCherry achieved by exciting the acceptor directly using 530 and 540 nm light, overlaid with the reported pure spectrum for the red FP in each condition. The average of each condition is shown next to the reported pure spectrum for each acceptor is shown in **(C)**, **(E)**, and **(G)** respectively. Scan of both mScarlet-I and mCherry in the NG-mScarlet-I and NG-mCherry constructs faithfully replicated the upstroke and peak of purified mScarlet-I and mCherry, with the major difference between the observed and pure protein spectrum being a faster decay of the tail of the spectrum at high wavelengths. In contrast, the scans of mRuby3 in the NG-mRuby3 construct revealed an emission spectrum that was 6 nm shifted from what was reported for purified mRuby3. Due to the differences seen with each of the red acceptor proteins and varying levels of background, custom acceptor emission spectrums were created to serve as the acceptor emission for linear umixing shown in **(C)**, **(E)**, and **(G)** as the black dashed line. The raw traces used to determine the efficiency of **(H)** NG-mRuby3, **(I)** NG-mScarlet-I, and **(J)** NG-mCherry are shown, corresponding to the efficiency graph in Fig 2F.
(TIF)

**S2 Fig. EGFP performance under 2-Photon FLIM Imaging. (A)** Lifetime data collected from individual HEK293 cells expressing cytosolic EGFP at various laser powers up to 25 W/cm$^2$ after 50 frames. Black bars indicate the average ± 95% confidence interval. **(B)** Lifetime and **(C)** intensity of samples taken over 300 frames at various laser powers. * = P < 0.05, ** = P< 0.005, and *** = P < 0.0005 compared to the frame matched 5W/cm$^2$ dataset. N for each sample is as follows 5W/cm$^2$: 10 cells, 10W/cm$^2$: 10 cells, 15W/cm$^2$: 10 cells, 20W/cm$^2$: 9 cells, 25W/cm$^2$: 10 cells.
(TIF)

**S3 Fig. Workflow for confocal imaging analysis.** Example workflow demonstrating how confocal images were processed to create intensity slope histograms in Fig 5.
(TIF)

**S4 Fig. Immunoblot of NG-Stop and NG-Red FP tandem constructs.** 10 μg of total protein derived from cell transiently transfected with the given construct one day post transfection was loaded into a 16% SDS-PAGE gel and mNeonGreen was visualized using an **(A)** anti-mNeonGreen antibody and **(B)** an anti-GFP (for the detection of mClover3 containing constructs). NG-Stop has presents a band near it's predicted molecular weight of 27kDa. Each of the tandems except NG-P2A-mRuby3, show a bright band at the full predicted weight of the tandem constructs, 54kDA. Importantly, only the NG-Stop and NG-P2A-mRuby3 show bands corresponding to a monomeric mNeonGreen. NG-mCherry, NG-mScarlet-I, NG-mScarlet, and mClover3-mRuby3 show products in between the expected full tandem and the mNeonGreen monomer that are likely due to the hydrolysis of the red FPs backbone during cell lysis and subsequent protein denaturation that has been reported previously with DsRed like red FPs[38]. Note that the band that appears between 75 and 100 kDa in **(A)** is present in all lanes indicating that it is a non-specific target of the mNeonGreen antibody.
(TIF)

**S5 Fig. mRuby3 performance as a FRET acceptor for the GFP like green-yellow FP mClover3. (A)** Confocal merge image of HEK293 cells expressing mClover3-mRuby3 demonstrates that mClover3-mRuby3 also displays high expression heterogeneity, similar to what was seen with other mRuby3 constructs. (B) Lifetimes of HEK293 cells expressing mClover3-Stop (the donor only condition) and mClover3-mRuby3. Each symbol represents a measurement from a single cell. Black bars indicate the average ± 95% confidence interval. N for each sample is as follows, mClover3-Stop: 65 cells and mClover3-mRuby3: 71 cells. *** = P < 0.0005 compared to mClover3-Stop. **(C)** The average FRET efficiency of mClover3-mRuby3.
(TIF)

**S6 Fig. Example decay curves from the NG-mRuby3 time series. (A)** Example fluorescence decay curves of a single cells expressing NG-mRuby3 2–5 days post transfection (DPT) representative of the average for each condition. The NG-Stop curve and NG-mRuby3 1 DPT curves from Fig 3C are also shown for reference. **(B)** Example fluorescence decay curves from a single cell expressing NG-mRuby3 5 DPT before and after acceptor photobleaching. The NG-Stop curve from Fig 3C is repeated here for reference.
(TIF)

**S1 Raw Images. Raw immunoblot data for S4 Fig.** Raw and unmodified immunoblot scans used in the construction of S4 Fig.
(PDF)

**S1 Table. Amino acid sequences of constructs used in this manuscript.** The color-coded amino acid sequences for each construct are shown above. For NG-P2A-mRuby3, the cleavage

site is found between the glycine and proline residues found immediately before the mRuby3 sequence and is indicated with a |.
(TIF)

## Acknowledgments

We would like to thank both David Yule and Andrew Wojtovich for helpful comments over the course of the study, as well as generous sharing of equipment.

## Author Contributions

**Conceptualization:** Tyler W. McCullock, David M. MacLean.

**Formal analysis:** Tyler W. McCullock.

**Funding acquisition:** David M. MacLean, Paul J. Kammermeier.

**Investigation:** Tyler W. McCullock.

**Methodology:** Tyler W. McCullock, David M. MacLean.

**Resources:** David M. MacLean, Paul J. Kammermeier.

**Supervision:** David M. MacLean, Paul J. Kammermeier.

**Visualization:** Tyler W. McCullock.

**Writing – original draft:** Tyler W. McCullock.

**Writing – review & editing:** David M. MacLean, Paul J. Kammermeier.

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
