## [Decision Letter · Decision Letter 0]

30 Aug 2019

PONE-D-19-18735

Comparing the performance of mScarlet-I, mRuby3, and mCherry as FRET acceptors for mNeonGreen

PLOS ONE

Dear Mr. McCullock,

Thank you for submitting your manuscript to PLOS ONE. After careful consideration, we feel that it has merit but does not fully meet PLOS ONE’s publication criteria as it currently stands. Therefore, we invite you to submit a revised version of the manuscript that addresses the points raised by reviewers during the review process.

All three reviewers have provided many valuable suggestions and I hope that you will be able to address them.

We would appreciate receiving your revised manuscript by Oct 14 2019 11:59PM. To enhance the reproducibility of your results, we recommend that if applicable you deposit your laboratory protocols in protocols.io, where a protocol can be assigned its own identifier (DOI) such that it can be cited independently in the future. For instructions see: http://journals.plos.org/plosone/s/submission-guidelines#loc-laboratory-protocols

We look forward to receiving your revised manuscript.

Kind regards,

Vadim E. Degtyar, Ph.D.

Academic Editor

PLOS ONE

Journal Requirements:

Reviewers' comments:

Reviewer's Responses to Questions

**Comments to the Author**

1. Is the manuscript technically sound, and do the data support the conclusions?

Reviewer #1: Yes

Reviewer #2: Yes

Reviewer #3: No

2. Has the statistical analysis been performed appropriately and rigorously? 

Reviewer #1: Yes

Reviewer #2: Yes

Reviewer #3: Yes

3. Have the authors made all data underlying the findings in their manuscript fully available?

Reviewer #1: Yes

Reviewer #2: Yes

Reviewer #3: Yes

4. Is the manuscript presented in an intelligible fashion and written in standard English?

Reviewer #1: Yes

Reviewer #2: Yes

Reviewer #3: Yes

5. Review Comments to the Author

Reviewer #1: Manuscript PONE-D-19-18735 from Kammermeier and coworkers describes a systematic comparison of 3 different RFPs (mRuby3, mScarlet, and mCherry) as FRET acceptors for an mNeonGreen donor FP. Systematic characterization efforts such as this are an important contribution to the scientific literature since they can save other labs time, money, and effort in picking the best FRET pair to use for their own investigations. Overall, the authors achieve exactly what they set out to do in a fairly straightforward manner. In addition, the manuscript is well-written, logically organized, and the results are presented very clearly. Essentially, the authors found that mScarlet and mCherry behaved pretty much as expected, but that mRuby3 exhibits a much lower FRET efficiency than expected. This is exactly the sort of unexpected result that shows why studies such as this are important. One way that the authors could have increased the impact of this paper would have been to also evaluate mNeonGreen use EGFP as FRET donors for these applications. Given the lack of previous studies on the performance of mNeonGreen under 2P illumination, it would have been helpful to perform side-by-side 2P characterization of brightness and lifetime with EGFP. Based on this study, the authors conclude that mNeonGreen is suitable. It would have been useful to know if this means it is better, worse, or similar to EGFP.

My recommendation is that this manuscript will be suitable for publication once the following comments and minor revisions have been addressed.

Comments and concerns to be addressed

1. The authors state that “...histograms of the slopes for individual cell expressing the NG-Cherry and NG-Scarlet constructs revealed a reasonable distribution and narrow spread of slopes (Fig 5C,D).” While it is true that both Cherry and Scarlet are narrower than Ruby, it does appear as though mCherry has a much narrower spread than mScarlet, and this would seem to represent a desirable property in a FRET acceptor. Given this, plus the fact that mCherry gave identical quenching to mScarlet in FLIM, suggests that (at least for FLIM) mCherry might actually be the better choice. This should be pointed out in the conclusion.

2. Most cells expressing NG-mRuby3 were observed to decrease their lifetime over the course of several days, suggesting that the low FRET efficiency observed 1 DPT was due to the slow maturation. Some cells were observed to still be exhibiting donor-only like lifetimes after 5 DPT. Perhaps a reasonable explanation for this result is cell-to-cell heterogeneity in terms of the rate of protein turnover? That is, cells that have faster turnover, and faster rate or protein synthesis, might be expected to never accumulate substantial amounts of the fully mature fusion protein, no matter how long they are cultured. One way to examine this might be to plot lifetime vs mNeonGreen fluorescence intensity. If this hypothesis is correct, brighter cells might be expected to have lower lifetimes due to slower turnover of the protein and accumulation to higher levels. In both the abstract and discussion, the authors emphasize that the reasons for the poor performance of mRuby3 are unclear. It does seem like, ultimately, it is the very slow maturation (and possibly efficiency of maturation and/or rate of protein turnover) that is the culprit. Based on the data in Table 1, that is the one characteristic in which mRuby3 is the outlier. I think that the authors could be bit more decisive in pinning the blame on this factor.

3. The explanation that short linkers (line 487) are the reason why mCherry and mScarlet performed similarly does not make sense. The observed FRET efficiencies are all in the 20-30% range, meaning that the two fluorophores are at distances greater than the Forster radii. It is only when two fluorophores get very close that FRET efficiencies approach 100% and differences in Forster radii become irrelevant. I suspect that the two factors that really matter here are the time for protein maturation as well as the fraction of RFP proteins that make a fluorophore. I suspect that mCherry is particularly efficient at forming the chromophore, which is why it typically provides a bright signal in cells, even though its quantum yield and extinction coefficient are not as high as other proteins.

4. I would like to suggest that the authors make it clear about the performance of these FPs for FRET (which only refers to donor quenching) and sensitized emission. For example, in the Conclusion the author’s state that “... but mScarlet-I outperformed mCherry in the intensiometric study due to its higher quantum yield.” It should be clear that the better performance is due to greater sensitized emission due to the higher quantum yield. Strictly speaking, the quantum yield of the acceptor is irrelevant to FRET (but critical for sensitized emission, of course).

Minor revisions

- To someone who is not an expert in this field, the term 'better tissue penetrance’ in reference to green and red fluorophores. This should be reworded to make it clear that longer wavelengths of excitation and emission light are what better penetrate through tissue.

- I think it is important to always include the ‘m’ and the ‘-I’ in the case of mScarlet-I, when referring to the FPs. Dropping these could cause some confusion for readers.

- Line 99: missing ‘hours’

- Line 234: its

- Line 295: We first sought...

Reviewer #2: This manuscript presents a useful evaluation of three red-emitting FRET acceptors for the green FP mNeonGreen. The results are presented clearly and the work was clearly rigorous.

Suggestions:

1. The authors should be very clear in the caveat that these measurements were done using only one configuration (N-terminal mNeonGreen, C-terminal red FPs, only one linker used), and may not be generalizable to other constructs or sensors made using these FPs. The work was also done using only one donor and three acceptors, and thus is not generalizable to any other potential FRET pairs.

2. The presence of a second, lower MW band on Western blots is not surprising when using red-emitting FPs with a DsRed-type chromophore, which all three acceptor FPs possess. This is due to hydrolysis of the red chromophore upon protein denaturation (the acylimine moiety is vulnerable to hydrolysis), and is generally not an indicator of proteolysis. The authors may wish to revise their discussion of this observation to reflect this.

3. Despite the fact that mRuby3 clearly does not perform well in this assay, it remains mysterious and concerning that the authors observed such extreme heterogeneity in their transfected cells even after several days of maturation time. From the description of the constructs, it seems possible that the same DNA sequence, encoding the amino acid sequence SKGEE, may appear three separate times in each construct, since all of the FPs being tested have "GFP ends," i.e. the first and last seven amino acids have been altered or appended to match those of GFP (MVSKGEE....GMDELYK). If it is indeed the case that this 15 base pair sequence is repeated three times in each construct, it is plausible that some recombination may occur after transfection. Ideally, the authors should fix this issue if it indeed exists.

Reviewer #3: In this manuscript, McCullock et al tested mNeonGreen fused with one of three RFPs (mCherry, mRuby3, and mScarlet) for FRET efficiency in 2 assays (1p emission unmixing and 2p FLIM), finding the FRET efficiency the an mNeonGreen-mRuby3 tandem fusion was not as high as that of the mCherry or mScarlet fusion. They also investigated the variability of maturation of each fluorophore within the fusion proteins, finding that more cells showed high green and low red fluorescence or vice versa when expressing the mRuby3 fusion than when expressing the mCherry or mScarlet fusion. They also find that the FRET efficiency (measured by FLIM) improves substantially between 1 day and 2 days post transfection of the mNeonGreen-mRuby3 fusion. These last two results is indicative of inconsistent folding of mNeonGreen or of mRuby3 within the mNeonGreen-mRuby3 fusion protein.

I praise the authors for performing a comparison between mCherry, mRuby3, and mScarlet which are perhaps the three most robustly performing RFPs. The finding that mRuby3 has high variability in maturation within one particular constructs, the mNeonGreen-mRuby3 fusion, while mCherry and mScarlet do not in similar constructs, certainly raises the caveat that one cannot always assume all protein domains can tolerate fusions very well. However this study falls short of convincingly proving its conclusions for the following reasons:

1. The conclusions of this paper are written to apply to all uses of FRET but the example chosen of a tandem fusion between acceptor and donor is actually unlike any FRET reporter (it is more conformationally restricted than any FRET reporter would be). What about a real FRET reporter such as a single-chain kinase reporter (the prototype being AKAR)? And what about bimolecular reporters such as those used extensively by Ryohei Yasuda using FLIM?

2. Even within the confines of studying a donor-acceptor pair, only a single fusion protein (the mNeonGreen-mRuby3 protein) is being studied. It is premature to generalize or make predictions outside of this single protein.

3. Even within the even more restricted topic of whether mNeonGreen and mRuby3 can show high FRET coupling within a tandem fusion, then some simple troubleshooting is required at a minimum. Whereas previous studies used mClover3 and mRuby3 with good results, this study has swapped mClover3 for mNeonGreen. The new fusion protein may very well not allow folding of mRuby3 as well as previous fusion proteins.

a. For example, the mNeonGreen polypeptide may interfere with the mRuby3 polypeptide during protein folding; this might be the case either due to slower folding of mNeonGreen or specific peptide sequences within prefolded mNeonGreen binding to sequences of prefolded mRuby3. It is well known that domains of proteins can interfere with each other during folding, and natural proteins have evolved mechanisms such as translational pauses using rare codons and stiff linkers to allow domains to fold more independently.

b. Another reason for poor folding could be in the linkers themselves. It is will known that the sequence of linkers can greatly affect the folding of proteins attached to the linkers, e.g. George and Heringa 2002, PMID 12538905, or Chen et al 2014, PMID 23026637. Proline-rich and glycine-rich linkers are less likely to themselves interfere with the folding of an attached domain. Based on the authors’ description of their fusion protein, the linker between folded elements of mNeonGreen and mRuby3 has the sequence VMGMDELYKVSKEE, where the VMGMDELYK is brought in from mNeonGreen. This sequence contains a large proportion of hydrophobic and charged residues, which could interact with mRuby3 sequences to intefere with folding, and a scarcity of glycine or proline residues.

4. The authors chose to test only mNeonGreen as a donor. However it is not that much brighter than mClover3 (QY of 80% vs 78%), and the mRuby3 used here was actually taken out of the mClover3-mRuby3 fusion which does show high FRET. It would seem that the mClover3-mRuby3 should be used as a positive control for FRET (the authors could just use the existing construct they obtained from Addgene).

5. Likewise the authors make no comparisons between mNeonGreen and other GFPs in fluorescence lifetime or brightness or photostability in their FLIM system. It was reported that mNeonGreen is as photostable as EGFP and more photostable than mClover3 in one-photon illumination, but relative photostability in two-photon illumination is unknown. The authors have already done the characterization for mNeonGreen, so it would be useful to characterize EGFP or mClover3 for comparison.

6. PLOS authors have the option to publish the peer review history of their article (what does this mean?). If published, this will include your full peer review and any attached files.

Reviewer #1: No

Reviewer #2: No

Reviewer #3: No

---

## [Author Response · Author response to Decision Letter 0]

2 Jan 2020

We would like to thank the editor and reviews for taking time to critically review our manuscript and for providing a valuable constructive critique of our work. We have read their comments carefully and have attempted to integrate their feedback into our revised manuscript to the best of our ability. This included the development and evaluation of several new constructs, as well as extensive text revisions within results and discussion sections. We will begin by addressing several comments that were common among more than one reviewer before addressing each reviewer’s remaining comments below.

General Comments

Reviewers #1 and #3 both requested that we provide lifetime and photobleaching data for a more common FP under our 2-photon imaging system to act as a reference for mNeonGreen’s performance.

We agree that repeating these experiments with an addition reference protein would be beneficial, so examined EGFP’s stability using the same methodology and report these findings in S2 Fig within the Supplemental Information as well as include a short discussion of the results with the results section. 

Reviewer #2 and #3 both expressed concern over the manuscript over-generalizing our results, stressing that we are testing our fluorescent proteins (FP) under a limited number of conditions which may not be representative of all possible FRET experiment using mNeonGreen (NG) and the various red FPs in our manuscript.

We would like to thank the reviewers for pointing this out, as we certainly agree that our results may not be representative of all NG-red FP FRET. To address this, we made text edits to our manuscript where appropriate to stress that we developed and are testing a model system. Furthermore, we include a new paragraph at the end of the Discussion section to highlight the parts of our findings that may be more generalizable to other systems, while discussing how there may be unpredictable differences between our systems and others, stressing that it’s important to conduct further testing within an individual’s system of choice. 

Reviewer #1

1. The authors state that “...histograms of the slopes for individual cell expressing the NG-Cherry and NG-Scarlet constructs revealed a reasonable distribution and narrow spread of slopes (Fig 5C,D).” While it is true that both Cherry and Scarlet are narrower than Ruby, it does appear as though mCherry has a much narrower spread than mScarlet, and this would seem to represent a desirable property in a FRET acceptor. Given this, plus the fact that mCherry gave identical quenching to mScarlet in FLIM, suggests that (at least for FLIM) mCherry might actually be the better choice. This should be pointed out in the conclusion.

Initially we were cautious of our interpretation of our slope data because it is unclear if mCherry’s narrower distribution was due to a higher level of expression/maturation efficiency or if its due to mCherry’s lower brightness, or both. That being said, the distribution of lifetimes we received for NG-mCherry was also narrower than that of NG-mScarlet-I, so we are willing to agree that mCherry did act as a more consistent acceptor and modified the text of the conclusion to include this.

2. Most cells expressing NG-mRuby3 were observed to decrease their lifetime over the course of several days, suggesting that the low FRET efficiency observed 1 DPT was due to the slow maturation. Some cells were observed to still be exhibiting donor-only like lifetimes after 5 DPT. Perhaps a reasonable explanation for this result is cell-to-cell heterogeneity in terms of the rate of protein turnover? That is, cells that have faster turnover, and faster rate or protein synthesis, might be expected to never accumulate substantial amounts of the fully mature fusion protein, no matter how long they are cultured. One way to examine this might be to plot lifetime vs mNeonGreen fluorescence intensity. If this hypothesis is correct, brighter cells might be expected to have lower lifetimes due to slower turnover of the protein and accumulation to higher levels. In both the abstract and discussion, the authors emphasize that the reasons for the poor performance of mRuby3 are unclear. It does seem like, ultimately, it is the very slow maturation (and possibly efficiency of maturation and/or rate of protein turnover) that is the culprit. Based on the data in Table 1, that is the one characteristic in which mRuby3 is the outlier. I think that the authors could be bit more decisive in pinning the blame on this factor.

We would like to thank the reviewer for the suggestion, as we do think it is likely that differences in protein turnover or stability from one cell to another causes the heterogeneity we observe with slow maturing acceptors (All mRuby3 constructs as well as a new mScarlet construct). Unfortunately, we are not able to conduct the suggested analysis due to our lifetime imaging methodology. While collecting lifetimes, it is important for us to keep the counts per second our detectors receive within a particular range to allow for accurate detection of lifetimes without overloading the detectors, which would cause artificial shorting of the lifetime (or damage to the detector in extreme situations). Because of this, we modulate the laser power used from field to field as necessary, which prevents us from making comparisons of cell intensity from one field to another. We did include a discussion of this idea within Discussion section though. 

3. The explanation that short linkers (line 487) are the reason why mCherry and mScarlet performed similarly does not make sense. The observed FRET efficiencies are all in the 20-30% range, meaning that the two fluorophores are at distances greater than the Forster radii. It is only when two fluorophores get very close that FRET efficiencies approach 100% and differences in Forster radii become irrelevant. I suspect that the two factors that really matter here are the time for protein maturation as well as the fraction of RFP proteins that make a fluorophore. I suspect that mCherry is particularly efficient at forming the chromophore, which is why it typically provides a bright signal in cells, even though its quantum yield and extinction coefficient are not as high as other proteins.

We would like to thank the reviewer for pointing out the error in our logic and have replaced the relevant discussion section with one suggesting this may be due to mCherry’s maturation properties.

 4. I would like to suggest that the authors make it clear about the performance of these FPs for FRET (which only refers to donor quenching) and sensitized emission. For example, in the Conclusion the author’s state that “... but mScarlet-I outperformed mCherry in the intensiometric study due to its higher quantum yield.” It should be clear that the better performance is due to greater sensitized emission due to the higher quantum yield. Strictly speaking, the quantum yield of the acceptor is irrelevant to FRET (but critical for sensitized emission, of course).

We have modified our conclusions to better clarify this point.

- To someone who is not an expert in this field, the term 'better tissue penetrance’ in reference to green and red fluorophores. This should be reworded to make it clear that longer wavelengths of excitation and emission light are what better penetrate through tissue.

We have added additional wording to clarify.

- I think it is important to always include the ‘m’ and the ‘-I’ in the case of mScarlet-I, when referring to the FPs. Dropping these could cause some confusion for readers.

We have modified the names of constructs throughout the paper (text and figures) to include the full names of the red FPs for clarity.

Reviewer #2

1. The authors should be very clear in the caveat that these measurements were done using only one configuration (N-terminal mNeonGreen, C-terminal red FPs, only one linker used), and may not be generalizable to other constructs or sensors made using these FPs. The work was also done using only one donor and three acceptors, and thus is not generalizable to any other potential FRET pairs.

Please see general comments above.

2. The presence of a second, lower MW band on Western blots is not surprising when using red-emitting FPs with a DsRed-type chromophore, which all three acceptor FPs possess. This is due to hydrolysis of the red chromophore upon protein denaturation (the acylimine moiety is vulnerable to hydrolysis), and is generally not an indicator of proteolysis. The authors may wish to revise their discussion of this observation to reflect this.

We would like to thank the review for pointing out that the acylimine moiety is vulnerable to hydrolysis. We have updated our discussion of the western blot results (which are now mostly contained within the S4 Figure caption) to specify that we were referring to proteolysis of the linker itself, and that extra bands between monomeric mNeonGreen/mClover3 and the dimer band are due to this process.

3. Despite the fact that mRuby3 clearly does not perform well in this assay, it remains mysterious and concerning that the authors observed such extreme heterogeneity in their transfected cells even after several days of maturation time. From the description of the constructs, it seems possible that the same DNA sequence, encoding the amino acid sequence SKGEE, may appear three separate times in each construct, since all of the FPs being tested have "GFP ends," i.e. the first and last seven amino acids have been altered or appended to match those of GFP (MVSKGEE....GMDELYK). If it is indeed the case that this 15 base pair sequence is repeated three times in each construct, it is plausible that some recombination may occur after transfection. Ideally, the authors should fix this issue if it indeed exists.

The reviewer is correct that this sequence does appear two or three times for all constructs. For clarity, we include a new supplemental table (S1 Table) to clearly disseminate the amino acid sequence of our constructs. As for recombination, we believe that it would be more likely to occur within bacteria while we were making or propagating the constructs, and we have assured that this is not happening through sanger sequencing of each construct before it was used in the mammalian cells. If recombination was to occur in the mammalian cells to a significant extent that resulted in mNeonGreen without a red protein acceptor, we should be able to detect that with western blotting, and we did not (S4 Fig). Given that data, and our confocal data which suggest that the major phenotype is mNeonGreen fluorescence without red FP fluorescence, we do not believe recombination within our HEK293 cells is a problem.

Reviewer #3

1. The conclusions of this paper are written to apply to all uses of FRET but the example chosen of a tandem fusion between acceptor and donor is actually unlike any FRET reporter (it is more conformationally restricted than any FRET reporter would be). What about a real FRET reporter such as a single-chain kinase reporter (the prototype being AKAR)? And what about bimolecular reporters such as those used extensively by Ryohei Yasuda using FLIM?

We hope that the information we disseminate in this manuscript will ultimately aid the development of new and exciting FRET systems of various kinds, including but not limited to biosensors. Although we are capable of testing these FPs outside of our model system, we do not believe there is a “standard” system that would enable fair comparison between the results we received with our model system and other systems, without testing each individual system. As reviewer #2 and #3 pointed out, the data we receive from our system is situational, but that assumption also applies to other systems as well. Given this, we do not feel like it will be an overly meaningful addition to develop and test a new set of reporters.

2. Even within the confines of studying a donor-acceptor pair, only a single fusion protein (the mNeonGreen-mRuby3 protein) is being studied. It is premature to generalize or make predictions outside of this single protein.

During the process of revising the manuscript, we developed several new constructs which are described below. 

3. Even within the even more restricted topic of whether mNeonGreen and mRuby3 can show high FRET coupling within a tandem fusion, then some simple troubleshooting is required at a minimum. Whereas previous studies used mClover3 and mRuby3 with good results, this study has swapped mClover3 for mNeonGreen. The new fusion protein may very well not allow folding of mRuby3 as well as previous fusion proteins.

a. For example, the mNeonGreen polypeptide may interfere with the mRuby3 polypeptide during protein folding; this might be the case either due to slower folding of mNeonGreen or specific peptide sequences within prefolded mNeonGreen binding to sequences of prefolded mRuby3. It is well known that domains of proteins can interfere with each other during folding, and natural proteins have evolved mechanisms such as translational pauses using rare codons and stiff linkers to allow domains to fold more independently.

b. Another reason for poor folding could be in the linkers themselves. It is will known that the sequence of linkers can greatly affect the folding of proteins attached to the linkers, e.g. George and Heringa 2002, PMID 12538905, or Chen et al 2014, PMID 23026637. Proline-rich and glycine-rich linkers are less likely to themselves interfere with the folding of an attached domain. Based on the authors’ description of their fusion protein, the linker between folded elements of mNeonGreen and mRuby3 has the sequence VMGMDELYKVSKEE, where the VMGMDELYK is brought in from mNeonGreen. This sequence contains a large proportion of hydrophobic and charged residues, which could interact with mRuby3 sequences to intefere with folding, and a scarcity of glycine or proline residues.

We thank the reviewer for this discussion, and we agree that simply because our model system worked well for mScarlet-I and mCherry does not mean it was necessarily a good system to evaluate mRuby3. The study that originally published mRuby3 (Bajar, et al (2016); ref #10) evaluated its performance as an acceptor for EGPF, mClover3, and mNeonGreen, and reported excellent results with each (35%, 45%, and 48% FRET efficiencies respectively). In that study they removed the “GFP ends” (GITHGMDELYK for EGFP and mClover3) or (GMDELYK for mNeonGreen) from the donors and fused them to mRuby3 (amino acids 3-233) with the following linker sequence: LESFFEDPMVSKGEE resulting in an effective linker of about 12 amino acids. Given this, we believe it’s is unlikely that an mNeonGreen which is 17 amino acids away would interfere with mRuby3 more than one that is approximately 12 amino acids away. Nonetheless, we developed two new tandems containing mRuby3: a construct with mNeonGreen fused to mRuby3’s c-terminus through GGSGG linker, and a construct in which a P2A self-splicing peptide was inserted between the mNeonGreen and mRuby3. Additionally, we tested a slow maturing variant of mScarlet-I, mScarlet. The data pertaining to these constructs are found in the new Fig 6, and lead us to conclude that factors intrinsic to mRuby3 are the source of its deficiencies as an acceptor, particularly its slow maturation. As for the composition of what we consider our “17 amino acid linker” in our first tandems (NG-mRuby3 and others), we chose to maintain the “GFP ends” of the proteins because we feel these are less frequently removed then they are kept. 

4. The authors chose to test only mNeonGreen as a donor. However it is not that much brighter than mClover3 (QY of 80% vs 78%), and the mRuby3 used here was actually taken out of the mClover3-mRuby3 fusion which does show high FRET. It would seem that the mClover3-mRuby3 should be used as a positive control for FRET (the authors could just use the existing construct they obtained from Addgene).

We tested the mClover3-mRuby3 performance and report the data in S5 Fig, as well as a description of the results in the Results section. In summary, our data from this construct was more consistent our mNeonGreen donor data than what was previously reported for mClover3 as a donor (Bajar, et al (2016); ref #10). 

5. Likewise the authors make no comparisons between mNeonGreen and other GFPs in fluorescence lifetime or brightness or photostability in their FLIM system. It was reported that mNeonGreen is as photostable as EGFP and more photostable than mClover3 in one-photon illumination, but relative photostability in two-photon illumination is unknown. The authors have already done the characterization for mNeonGreen, so it would be useful to characterize EGFP or mClover3 for comparison.

Please see general comments above.

---

## [Decision Letter · Decision Letter 1]

15 Jan 2020

PONE-D-19-18735R1

Comparing the performance of mScarlet-I, mRuby3, and mCherry as FRET acceptors for mNeonGreen

PLOS ONE

Dear Mr. McCullock,

Thank you for submitting your manuscript to PLOS ONE.  We are pleased to inform you that your manuscript has been judged scientifically suitable for publication, but still does not fully meet PLOS ONE’s publication criteria as it currently stands. Therefore, we invite you to submit a revised version of the manuscript that addresses a few critical points raised by Reviewers #1 and #3.

We would appreciate receiving your revised manuscript by Feb 29 2020 11:59PM, although it seems that required changes wouldn't take much time.  If you adequately address these minor suggestions, the manuscript may not require another round of review.  To enhance the reproducibility of your results, we recommend that if applicable you deposit your laboratory protocols in protocols.io, where a protocol can be assigned its own identifier (DOI) such that it can be cited independently in the future. For instructions see: http://journals.plos.org/plosone/s/submission-guidelines#loc-laboratory-protocols

We look forward to receiving your revised manuscript.

Kind regards,

Vadim E. Degtyar, Ph.D.

Academic Editor

PLOS ONE

Reviewers' comments:

Reviewer's Responses to Questions

**Comments to the Author**

1. If the authors have adequately addressed your comments raised in a previous round of review and you feel that this manuscript is now acceptable for publication, you may indicate that here to bypass the “Comments to the Author” section, enter your conflict of interest statement in the “Confidential to Editor” section, and submit your "Accept" recommendation.

Reviewer #1: All comments have been addressed

Reviewer #3: (No Response)

2. Is the manuscript technically sound, and do the data support the conclusions?

Reviewer #1: Yes

Reviewer #3: Yes

3. Has the statistical analysis been performed appropriately and rigorously? 

Reviewer #1: I Don't Know

Reviewer #3: Yes

4. Have the authors made all data underlying the findings in their manuscript fully available?

Reviewer #1: Yes

Reviewer #3: No

5. Is the manuscript presented in an intelligible fashion and written in standard English?

Reviewer #1: Yes

Reviewer #3: Yes

6. Review Comments to the Author

Reviewer #1: My initial recommendation was that this manuscript will be suitable for publication once my comments and minor revisions were addressed. The authors have thoroughly addressed the comments, though there are still some minor issues that will be need to be addressed. The references to Supplementary Figure 2 appear to be incorrect on lines 153, 156, and 212 (probably these should be to Supplementary Figures S1 and S3). The authors should carefully check the manuscript to make sure all of the figure references are correct, and consistently use "Fig. S2" rather than "S2 Fig" (as an example).

I would have liked to see the lifetime data for EGFP (Fig S2) with the data in Fig. 3ABC.

The word 'stable' is misspelled on line 330.

Reviewer #3: The authors have answered my major questions. I only suggest that they now add mClover3 and mScarlet to Table 1 ("Properties of the fluorescent proteins used in this study")

In addition, the slightly higher lifetime of mClover3 vs mNeonGreen would suggest it should be an equivalent (or maybe slightly better) FRET donor for mScarlet-I. The discussion states "mNeonGreen is an ideal donor for Green/Red FRET experiments. Its yellow shifted excitation and emission spectrum allow for a high degree of overlap with red FPs while also being capable of being excited with lower energy blue light than cyan or more blue shifted green FPs." It might be appropriate to include mClover3 in this statement as well.

7. PLOS authors have the option to publish the peer review history of their article (what does this mean?). If published, this will include your full peer review and any attached files.

Reviewer #1: No

Reviewer #3: No

---

## [Author Response · Author response to Decision Letter 1]

21 Jan 2020

We again thank the editor and reviewers for their time, comments and thoughts regarding our manuscript. We have corrected typographical errors and ensured all figure references are to the correct figure, and that all references are consistently formatted to PLOS standards. The remaining points are addressed below.

Reviewer #1: I would have liked to see the lifetime data for EGFP (Fig S2) with the data in Fig. 3ABC.

The overall goal of our work is compare various red acceptors. The specific goal of Figure 3 is to establish that mNG is a good donor for the 2-photon FLIM experiments with which we will compare various red acceptors, not that mNG is superior to other specific donors such as EGFP. Therefore, we believe the supplemental data is an appropriate place for our EGFP data as more expert or multi-photon familiar readers can refer to that section for the comparison of donors while the general reader interested in acceptor behavior can move to the next section. We would like to point out that we took care to design Figure S2 to permit ease of comparison of mNG in Figure 3 and EGFP in S2 (keeping the scaling in the case of panel A and the axes for panels C and D that same, as well as the color coding and symbols used, and significance indicators) despite the data being in two separate places in the manuscript. 

Reviewer #3: I only suggest that they now add mClover3 and mScarlet to Table 1 ("Properties of the fluorescent proteins used in this study")

We have added mClover3 and mScarlet to Table 1.

In addition, the slightly higher lifetime of mClover3 vs mNeonGreen would suggest it should be an equivalent (or maybe slightly better) FRET donor for mScarlet-I. The discussion states "mNeonGreen is an ideal donor for Green/Red FRET experiments. Its yellow shifted excitation and emission spectrum allow for a high degree of overlap with red FPs while also being capable of being excited with lower energy blue light than cyan or more blue shifted green FPs." It might be appropriate to include mClover3 in this statement as well.

We agree with Reviewer #3 that mClover3’s published properties would also suggest it would also make a suitable donor, similar to mNeonGreen. Unfortunately for the purposes of the manuscript, we did not extensively test mClover3 ourselves, and are therefore not comfortable making conclusions regarding how it may act as a donor and in Green/Red FRET experiments.

---

## [Editor Report · Decision Letter 2]

24 Jan 2020

Comparing the performance of mScarlet-I, mRuby3, and mCherry as FRET acceptors for mNeonGreen

PONE-D-19-18735R2

Dear Dr. McCullock,

We are pleased to inform you that your manuscript has been judged scientifically suitable for publication and will be formally accepted for publication once it complies with all outstanding technical requirements.

With kind regards,

Vadim E. Degtyar, Ph.D.

Academic Editor

PLOS ONE
---

## [Editor Report · Acceptance letter]

29 Jan 2020

PONE-D-19-18735R2 

Comparing the performance of mScarlet-I, mRuby3, and mCherry as FRET acceptors for mNeonGreen 

Dear Dr. McCullock:

I am pleased to inform you that your manuscript has been deemed suitable for publication in PLOS ONE. Congratulations! Your manuscript is now with our production department. 

With kind regards,

on behalf of

Dr. Vadim E. Degtyar 

Academic Editor

PLOS ONE